



# Reconstructing ocean subsurface salinity at high resolution using a machine learning approach

Tian Tian[1,2], Lijing Cheng[2,3], Gongjie Wang[4], John Abraham[5], Shihe Ren[6], Jiang Zhu[2], Junqiang Song[1], Hongze Leng[1]

[1]College of Meteorology and Oceanography, National University of Defense Technology, Changsha, 410073, China
[2]Institute of Atmospheric Physics, Chinese Academy of Sciences, Beijing,100029, China
[3]Center for Ocean Mega-Science, Chinese Academy of Sciences, Qingdao, 266071, China
[4]National Climate Center, Chinese Meteorological Administration, Beijing, 100081, China
[5]School of Engineering, University of St. Thomas, Minneapolis, 55105, MN, U.S.A
[6]Key Laboratory of Research on Marine Hazards Forecasting, National Marine Environmental Forecasting Center, Ministry of Natural Resources, Beijing, 100081, China

*Correspondence to*: Lijing Cheng (chenglij@mail.iap.ac.cn)

**Abstract.** A gridded ocean subsurface salinity dataset with global coverage is useful for research on climate change and its variability. Here, we explore a machine learning approach to reconstruct a high-resolution (0.25° × 0.25°) ocean subsurface (0–2000 m) salinity dataset for the period 1993–2018 by merging *in situ* salinity profile observations with high-resolution (0.25° × 0.25°) satellite remote sensing altimetry absolute dynamic topography (ADT), sea surface temperature (SST), sea surface wind (SSW) field data, and a coarse resolution (1° × 1°) gridded salinity product. We show that the feed-forward
neural network approach can effectively transfer small-scale spatial variations in ADT, SST and SSW fields into the 0.25° × 0.25° salinity field. The root-mean-square error (RMSE) can be reduced by ~11% on a global-average basis compared with the 1° × 1° salinity gridded field. The reduction in RMSE is much larger in the upper ocean than the deep ocean, because of stronger mesoscale variations in the upper layers. Besides, the new 0.25° × 0.25° reconstruction shows more realistic spatial signals in the regions with strong mesoscale variations, e.g., the Gulf Stream, Kuroshio, and Antarctic Circumpolar Current
regions, than the 1° × 1° resolution product, indicating the efficiency of the machine learning approach in bringing satellite observations together with *in situ* observations. The large-scale salinity patterns from 0.25° × 0.25° data are consistent with the 1° × 1°gridded salinity field, suggesting the persistence of the large-scale signals in the high-resolution reconstruction. The successful application of machine learning in this study provides an alternative approach for ocean and climate data reconstruction that can complement the existing data assimilation and objective analysis methods. The reconstructed
IAP0.25° dataset is freely available at http://dx.doi.org/10.12157/IOCAS.20220711.001 (Tian et al., 2022).





## 1 Introduction

Gridded ocean datasets with complete global ocean coverage are of great importance to marine and climate research(Lyman and Johnson, 2014; Durack et al., 2014; Ciais et al., 2013; Domingues et al., 2008; Bagnell and DeVries, 2021). For example, most climate monitoring applications depend on gridded products (Abram et al., 2019; Ishii et al., 2017; Liang et al., 2021).

As one of the most important physical parameters of sea water, salinity is a critical regulator for ocean circulation and a central indicator of the global water cycle (Durack, 2015; Durack et al., 2012; Skliris et al., 2014, 2020; Zika et al., 2018). However, *in situ* salinity observation data are sparse owing to the limitations of observation techniques, which brings difficulties to the generation of ocean salinity data products (Roemmich et al., 2019, 2009).

Currently, the estimation of salinity fields mainly relies on two approaches: (1) objective analysis methods based on *in situ*

observations, resulting in so-called objective analysis data products (Gaillard et al., 2016; Hosoda et al., 2008; Roemmich and Gilson, 2009; Cheng and Zhu, 2016; Lu et al., 2020); and (2) data assimilation methods, which combine numerical simulations and observations to result in what is referred to as a reanalysis product (Carton et al., 2018; Forget et al., 2015; Jean-Michel et al., 2021; Balmaseda et al., 2013). The accuracy of the more traditional objective analysis approach is critically dependent on the data coverage and the reliability of spatial covariance, which defines how the information is

propagated from data-rich to data-sparse regions (Von Schuckmann et al., 2014; Zhou et al., 2004). Previously available objective analysis products mostly have a horizontal resolution of $1° \times 1°$. Some examples include the products from the Institute of Atmospheric Physics (IAP), China (Cheng and Zhu, 2016; Cheng et al., 2020), the EN4 data from the UK Met Office (Good et al., 2013), the World Ocean Atlas from the National Center for Environment Information, National Oceanic and Atmospheric Administration (NOAA) (Zweng et al., 2019), and the Ishii data from the Japan Meteorological Agency

(Ishii and Kimoto, 2009). Previous assessments suggest the existence of spurious shifts during observation system transfer periods (i.e. ~1993, 2005) (Meyssignac et al., 2019; Palmer et al., 2017), indicating the challenge of this approach. The reanalysis approach relies on model simulations, which uses data assimilation schemes to constrain models with various types of observations, such as *in situ* and satellite remote sensing data. (Storto et al., 2019; Jean-Michel et al., 2021). Reanalysis product can of course be strongly impacted by inherent model biases, especially below the ocean subsurface

where significant model biases persist (Storto et al., 2019; Balmaseda et al., 2015). For example, the US Navy started to build the Modular Ocean Data Assimilation System (Fox et al., 2002) in the 1980s and combined it with an ocean numerical model to monitor the 3D temperature and salinity fields in real time and provide underwater environment information. However, Palmer et al. (2017) and Cheng et al. (2020) indicated that such reanalysis products have much larger spread than observational products for ocean heat content (temperature) and ocean salinity, suggesting caution when adopting the data

assimilation approach.

As higher resolution data are crucial for understanding ocean processes at multiple scales, such as the meso- and sub-mesoscale (McWilliams, 2016), a dataset with a resolution higher than $1° \times 1°$ and with sufficient data quality could be useful for ocean and climate research, e.g., resolving the vertical structure of warm and cold eddies. To reconstruct a dataset



with resolution higher than $1° \times 1°$, remote sensing observations are essential, as they can be used to incorporate smaller-scale signals that are insufficiently sampled by *in situ* salinity profile observations. Two types of approaches have been used previously to propagate surface information to the subsurface (McWilliams, 2016): dynamical and statistical. The dynamical approach adopts physical relationships between surface and subsurface variables. For example, Wang et al. (2013) proposed an isQG (interior + surface quasi geostrophy) method, using information such as sea surface height, sea surface temperature, and sea surface salinity (SSS) to invert the density and velocity fields of the ocean interior. The statistical approach, on the other hand, utilizes historical data to establish a statistical relationship between remote sensing observations and subsurface observations (e.g., linear regression between surface height and subsurface temperature/salinity). Some of the available statistical relationships have been derived by multiple linear regression models (Willis et al., 2003; Fox et al., 2002), empirical orthogonal functions (Lorenz, 1956; Hannachi et al., 2007), and the Gravest Empirical Modes method (Watts et al., 2001). Notably, however, both dynamical and statistical approaches are overly dependent on physical assumptions, are always simplified, and have important limitations. For example, the surface dynamic height is apparently nonlinearly correlated with subsurface temperature/salinity.

This study takes advantage of machine learning to develop an alternative approach to reconstructing high-resolution ($0.25° \times 0.25°$) ocean subsurface fields (i.e., salinity). Compared with traditional reconstruction methods, machine learning approaches do not rely on any simplified assumptions in the process of data reconstruction, and are able to learn and fit parameters automatically. There have been some attempts recently; for example, Wang et al. (2021) used remote sensing data and neural network methods to estimate the subsurface temperature in the western Pacific. Combining satellite data and Argo gridded products, Su et al. (2020) used neural networks to estimate ocean heat content anomalies over four different depths down to 2000 m. Lu et al. (2019) estimated the subsurface temperature through a clustering neural network method based on sea surface temperature, sea surface height, and wind field measurement data. They used the gridded monthly Argo data as the ground truth for training and validation. Although these studies provide some hints that machine learning approaches can be useful in data reconstruction applications, there are major deficiencies. First, some studies (Lu et al. 2019; Su et al. 2020; Wang et al. 2021) used Argo gridded data rather than *in situ* salinity observations data as the "truth" to train the machine learning model, and thus the reconstruction error in the Argo gridded data is embedded in the final reconstruction. Second, the spatial resolution of the reconstructed data is still $1° \times 1°$. The added value of high-resolution remote sensing data is not maximized; for example, the resultant $1° \times 1°$ field is still insufficient to resolve mesoscale signals. Finally, previous reconstructions have tended to focus mainly on temperature rather than salinity, and a comprehensive assessment of the uncertainty has always been absent. Besides, due to the lack of interpretability of machine learning approach, a thorough comparative evaluation with traditional methods is clearly a crucial line of study.

This paper explores the possibility of using a machine learning approach in which *in situ* observations and remote sensing data are merged to reconstruct the salinity changes at a $0.25° \times 0.25°$ horizontal resolution and a monthly temporal resolution from the surface down to a depth of 2000 m. The first objective is to understand how well the machine learning approach works for data reconstruction using the ocean salinity field as an example. Second, we produce a high-resolution salinity



dataset (0.25° × 0.25°) for the community, including a comprehensive evaluation and uncertainty estimates. The added value of remote sensing data in a high-resolution salinity reconstruction is demonstrated and discussed.

The rest of the paper is organized as follows: The data and methods employed in our study are presented in section 2. The performance of the dataset in terms of geographical pattern, regional analysis, and overall reconstruction performance is assessed in section 3. The uncertainty of the machine learning approach is examined in section 4. An analysis of the major climatic patterns is conducted in section 5. And finally, the results of the study are summarized and discussed in section 6.

## 2 Data and Methods

### 2.1 Input Data

#### 2.1.1 Remote sensing and IAP1° observational data

High-resolution satellite remote sensing data and IAP1° salinity data were used as inputs to train a machine learning model. The high-resolution data included sea surface absolute dynamic topography (ADT), sea surface temperature (SST), and sea surface wind (SSW) fields, which included zonal (USSW) and meridional (VSSW) components.

The ADT data were from the European Copernicus Marine Environment Monitoring Service (CMEMS), with altimeter data from several satellites (including Jason-3, Sentinel-3A, HY-2A, Saral/AltiKa, Cryosat-2, Jason-2, Jason-1, T/P, ENVISAT, GFO, and ERS1/2). An optimal interpolation was made when merging all the satellite data in order to compute gridded ADT information (Mertz et al., 2016). The dataset has a spatial resolution of 0.25° × 0.25° and a daily temporal resolution covering the period from January 1993 to December 2020. In this study, monthly averages over daily data were used.

The SST data were from the daily Optimum Interpolation Sea Surface Temperature (OISST) v2.1 data from NOAA. OISST is produced by interpolating and extrapolating SST observations from different sources, resulting in a smoothed field with complete global ocean coverage. The sources of data are satellite (Advanced Very High Resolution Radiometer) and *in situ* platforms (i.e., ships and buoys) (Banzon et al., 2016; Reynolds et al., 2007). Data are currently available from 1 September 1981 to the present day, with a spatial resolution of 0.25° × 0.25° and a daily temporal resolution. Monthly averaging of 120 daily data was required for this study.

The SSW data were from the Cross-Calibrated Multi-Platform (CCMP) V2 monthly dataset (Wentz et al., 2016). The processing of CCMP V2 now combines Version-7 Remote Sensing Systems radiometer wind speeds, QuikSCAT and ASCAT scatterometer wind vectors, moored buoy wind data, and ERA-Interim model wind fields using a variational analysis method to produce 0.25° × 0.25° gridded vector winds (Atlas et al., 2011). The dataset covers the period from July 125 1987 to April 2019.

The IAP1° salinity data give a global coverage of the oceans at a horizontal resolution of 1° × 1° in 41 vertical levels from a depth of 1 m down to 2000 m, and a monthly temporal resolution from 1940 to the present day (Cheng and Zhu, 2016; Cheng et al., 2017). This product combines *in situ* salinity profiles with coupled model simulations (from CMIP5) to derive



an objective analysis with the Ensemble Optimal Interpolation approach. The product is designed to minimize the sampling
error in representing large-scale and long-term climate changes and variabilities.

All of the above-mentioned products were interpolated into unified monthly and 0.25° × 0.25° spatial resolution fields,
which were used as inputs for our machine learning approach.

### 2.1.2 *In situ* salinity observations

*In situ* ocean salinity observations were sourced from the World Ocean Database (WOD) (Boyer et al., 2018). Data from all
available instruments [i.e., Argo (Li et al., 2017), Bottle, conductivity–temperature–depth (CTD)] were used in this study.
Quality flags from WOD were applied (only "good data" with a flag equal to zero were used). All of the profiles were first
interpolated to a standard 41 vertical levels between 1 and 2000 meters, as in the IAP1° data. Then, 12 monthly
climatologies were constructed using all data from 1990 to 2010 (centered around 2000), and all salinity profiles are
subtracting the monthly climatology to derive the anomaly profiles. The reconstructions are applied as anomaly fields, as in
previous objective analyses, because of the larger spatial decorrelation length scale than the original fields (i.e., Levitus et al.
2009; Cheng et al. 2017). Finally, the salinity anomalies were averaged into 0.25° × 0.25°, 1-month, and 41-level grid boxes
via simple arithmetic averaging. The gridded averaged salinity anomalies were regarded as the "truth" to train the machine
learning approach during the reconstruction. The gridded averages were used instead of the raw profiles because they were
able to minimize the spatial heterogeneity of the raw profiles, reduce the subgrid (<0.25°) noise, and improve the
computational efficiency. Through certain tests (not shown here), we did not find a better performance when using the raw
profiles compared to the gridded fields.

### 2.2 Independent Ocean products for evaluation and comparison

Three independent ocean products were used to assess the quality of the newly reconstructed high-resolution salinity data in
this study (hereafter denoted as IAP0.25°), including ARMOR3D data, SMAP satellite data, and EN4 gridded data.

The ARMOR3D V4 dataset is from CMEMS, which is a global three-dimensional temperature and salinity dataset with high
horizonal (0.25° × 0.25°) spatial resolution from 0–5500 m (Guinehut et al., 2012; Mulet et al., 2012). It combines satellite
observation data (ADT, geostrophic surface currents, SST) and *in situ* temperature and salinity profile data through a
multivariate/simple linear regression and an optimal interpolation method. This product offers four versions with different
temporal resolutions: near-real-time weekly data, near-real-time monthly data, multi-year reprocessed weekly data, and
multi-year reprocessed monthly data. The multi-year reprocessed monthly data were used in this study, which have a spatial
resolution of 0.25° × 0.25° and cover the period from January 1993 to December 2020. The spatial coverage ranges from
82.125°S to 89.875°N and 0.125°E to 359.875°E.



The SMAP V4 dataset is provided by NOAA, and comprises monthly 0.25° × 0.25° resolution SSS data. The SMAP data are
based on satellite observations, and have the capability to monitor the global SSS field in near real time, with mesoscale
information resolved for salinity (Vinogradova et al., 2019).

The EN4 gridded data were obtained from the UK Met Office. The EN4 dataset uses an optimal interpolation method to
construct the ocean subsurface gridded salinity field based on *in situ* salinity profile data. We chose the EN4-GR10 version
(Gouretski and Reseghetti, 2010), which has a spatial resolution of 1° × 1° and 42 vertical levels from 5 to 5500 m, and
temporal coverage from 1940 to 2018 (Li et al. 2019).

**Table 1. A list of the input datasets and validation products used in this study.**

| Data type | Dataset | Data source | Horizontal resolution | Vertical coverage and resolution | Time period |
|---|---|---|---|---|---|
| Input | ADT (Mertz et al., 2016) | CMEMS | 0.25° × 0.25° | Sea surface | 1993–2020 |
| Input | SST (Berrar, 2018) | OISST | 0.25° × 0.25° | Sea surface | 1981–2022 |
| Input | SSW (Wentz et al., 2016) | CCMP | 0.25° × 0.25° | Sea surface | 1987–2019 |
| Input | IAP1° (Cheng and Zhu, 2016; Cheng et al., 2017) | IAP | 1° × 1° | 41 levels (1–2000 m) | 1960–2021 |
| Input | Salinity observations (Boyer et al., 2018) | WOD | Averaged into 0.25° × 0.25° | Interpolated to 41 levels (2000m) | 1960–2021 |
| Validation | ARMOR3D (Mertz et al., 2016) | CMEMS | 0.25° × 0.25° | 50 levels (1–5000 m) | 1993–2020 |
| Validation | SMAP (Vinogradova et al., 2019) | NOAA | 0.25° × 0.25° | Sea surface | 2015–2019 |
| Validation | EN4 (Vinogradova et al., 2019) | UK Met Office | 1° × 1° | 42 levels (1–5500 m) | 1940–2018 |

## 2.3 Methods

### 2.3.1 Feed-forward neural network

All of the input parameters of the model training were converted to anomalies by subtracting their respective climatologies
within 1993–2015. For example, the input data were the IAP1° salinity anomalies (IAP1SA), the absolute dynamic
topography anomalies (ADTA), sea surface temperature anomalies (SSTA), and sea surface wind anomalies (SSWA), which
included zonal and meridional anomalies (USSWA/VSSWA). The standardization of data can improve the convergence
speed of the neural network method (LeCun et al., 2012). The method  proposed by Denvil-Sommer et al. (2019) was used to
perform sine/cosine transformations of latitude, longitude, and time. The Z-score standardization method was used to process
the ADTA, SSTA, USSWA, VSSWA, IAP1SA, and standard layer depth parameters.

The feed-forward neural network (FFNN) was used to reconstruct the subsurface salinity anomalies in this study (illustrated
in Fig. 1). An FFNN is a one-way, multi-layer structure network that includes an input layer, hidden layers, and an output





layer (Abdar et al., 2021; Contractor and Roughan, 2021; Gabella, 2021; LeCun et al., 2012; Moussa et al., 2016; Vikas Gupta, 2017). The zero layer is called the input layer, the last layer is called the output layer, and the other intermediate

layers are called the hidden layers. The neurons in the FFNN are arranged in layers, with each neuron belonging to a different layer. The neurons of each layer are fully connected; that is, each neuron is connected to the neurons of the previous layer and the next layer. The information in the network will only flow from the input layer, to the hidden layers, and then to the output layer; that is, the output of the previous layer is used as the input of the next layer, and the information of the next layer has no effect on the previous layer. Each layer of the FFNN is equivalent to a function, and the connection of the multi-

layer network is equivalent to a composite function to form a linear or nonlinear mapping from input variables to output variables. The complexity of the network depends on the number of neurons in each layer and the number of hidden layers. The more layers, the more complex the FFNN.

The structure of the FFNN used in this study is illustrated in Fig. 1, in which $x$ is the input parameter and the gridded salinity anomalies are regarded as "truth values" (denoted as $y$). The input $x$ can be expressed as follows:

$x$ = (longitude, latitude, time, depth, IAP1SA, ADTA, SSTA, USSWA, VSSWA),

where longitude, latitude, time and depth in the FFNN input layer are the latitude, longitude, observation month and standard layer depth for the salinity observation, respectively.

A training/testing approach was adopted to determine the key parameters and settings for the FFNN. The full salinity dataset was randomly divided into a training set (80% of the whole dataset) and a test set (20% of the whole dataset). The training

set was used to fit an FFNN algorithm, and the test set was used for performance evaluation. The root-mean-square error (RMSE) was calculated between the reconstruction based on the training set and the test dataset as a model evaluation metric. Once the RMSE begins to increase, the training iteration stops and the best parameters and settings are stored. A grid search strategy (Liashchynskyi et al. 2019) was used to optimize the structure of the neural network, and we found that the overall reconstruction effect was best when the neural network structure was composed of one input layer, one output layer, and four

hidden layers; the number of neurons in each hidden layer was set to 256, 128, 64, and 32; the activation function was the Rectified Linear Unit; and the learning rate was 0.001. Bayesian regularization (Foresee and Hagan. 1997) and dropout was used during model training to maintain generalization and prevent overfitting. Since the Bayesian regularization algorithm does not require cross-validation to ensure generalization, no validation set was defined in this process (Lu et al., 2019; Wang et al., 2021). Adopting these settings and parameters, the final FFNN model used for reconstruction was trained by the

full salinity dataset to ensure the best performance.



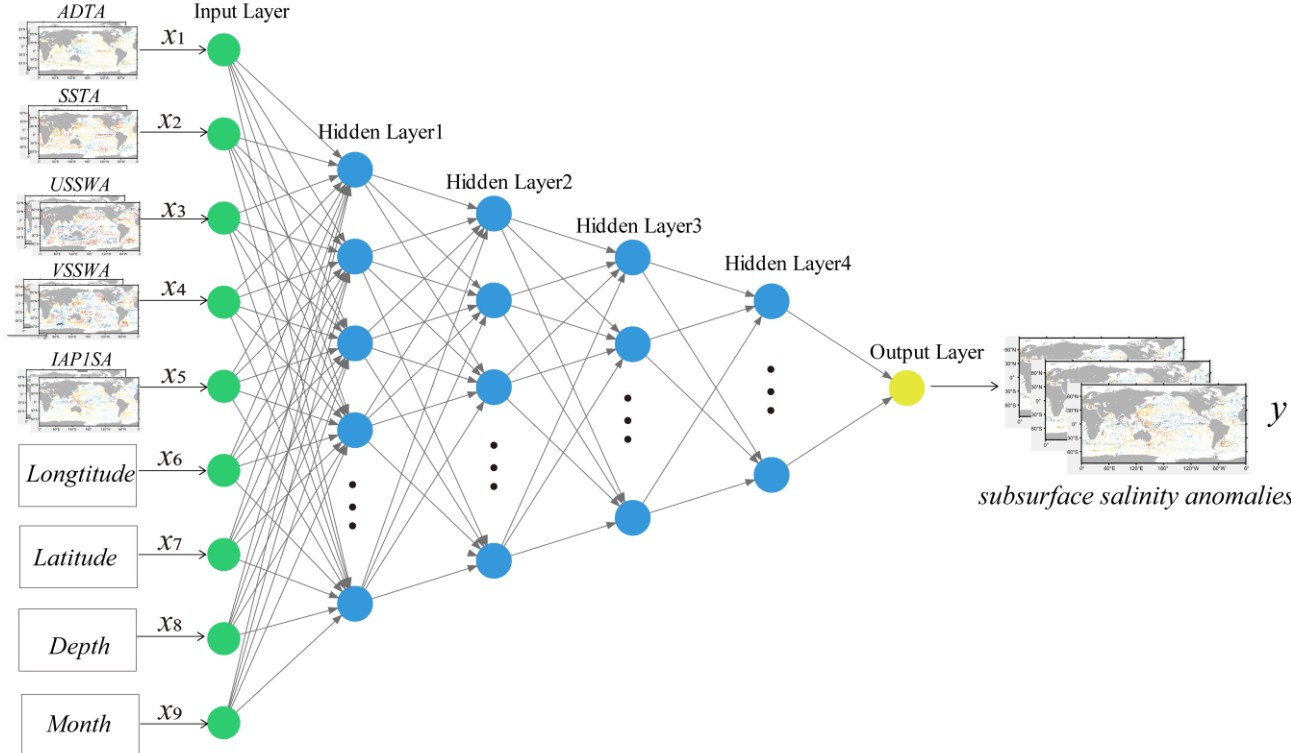

**Figure 1: Schematic diagram of subsurface salinity reconstruction by the FFNN approach.**

## 2.3.2 Five-fold cross validation

To evaluate the reconstruction using independent data, a 5-fold cross validation approach was used (Gui et al., 2020). This type of validation is one of the most popular models in statistics and machine learning (Berrar, 2018; Lei, 2020). In the five-fold cross validation, the full observational datasets were split into training (80%) and testing (20%) datasets. The training set was used to train the machine learning model and then to derive the reconstruction. The testing dataset was used as an independent dataset to verify the performance of the reconstruction. To effectively construct the training and testing datasets, the full observational data $0.25° \times 0.25°$ gridded average fields, as introduced before, were randomly split into five subsets. In each run, four subsets were used for training, and the remaining subset was used for testing. This process was repeated four times so that each of the five subsets could be used as the testing dataset. In this way, the five-fold cross validation needed to be trained five times to ensure that all the data participated in both the training and testing. By calculating the difference between the reconstructed data and the truth value of the independent testing data, the machine learning method could be evaluated.



### 2.3.3 Uncertainty quantification

The uncertainty of the final reconstruction will stem from three sources: instrumental error, representativeness error, and reconstruction error. The instrumental error is related to the instrument accuracy of a salinity measurement. For instance, the accuracy of most Argo salinity data is about ±0.01 psu. However, many Argo floats suffer from sensor drift, so the error is much larger for some data, which has yet to be quantified (Wong et al., 2020). For the validated marine mammal data, the accuracy can be ±0.03 psu (Siegelman et al., 2019). The CTD data accuracy is about ±0.01 psu. The representativeness error defines the accuracy of the gridded average in representing the true average of salinity in this grid. This error is associated with the sampling of ocean changes inside each 0.25° and 1-month grid: insufficient sampling will lead to errors in the grid average. The magnitude of error depends on the spatiotemporal variability in each grid box: larger subgrid variability requires more data to accurately estimate the grid average. The reconstruction error defines the accuracy of the gap-filling, i.e., the machine learning approach in this study. This error is related to the errors in the FFNN model's design and parameter choice, etc. To quantify the uncertainty in the reconstruction, these three major error sources had to be properly accounted for.

This study adopted a probabilistic approach to the propagation of errors and to quantify the uncertainty. We perturbed the interpolated 0.25° gridded average fields five times with prescribed instrumental and representativeness uncertainty, and then obtained five perturbed gridded fields. The FFNN was then applied to these five perturbed fields, separately, to result in five reconstructions at each time with a Monte Carlo dropout approach (Gal and Ghahramani, 2016; Abdar et al., 2021). This process led to 25 ensemble members in the final reconstruction, the spread of which was used to quantify the uncertainty.

The perturbations were performed as follows. For instrumental error, we perturbed each individual salinity profile by its instrumental accuracy, assuming a Gaussian distribution with zero mean and the instrumental accuracy as the standard deviation. The perturbed profiles were then used to construct the gridded average fields (0.25° and 1-month grid). On this basis, the representativeness error was included by perturbing each gridded average by a Gaussian distribution with zero mean and standard deviation of $Variance/\sqrt{n}$, where $Variance$ is a measure of subgrid variability (<0.25° and <1-month) and $n$ is the number of profiles to calculate the grid mean. The $Variance$ is static (non-time-varying) and calculated by the standard deviation of the salinity data in each 1° grid box when there were >10 measurements. Grid boxes of 1° were used here because of the data scarcity, which would overestimate the subgrid variability of <0.25°. However, this overestimation could compensate for the unresolved/unknown errors, such as salinity drift. The results are presented in Figs. 2a and c. The figures reveal incomplete ocean coverage. Thus, we interpolated this field using an objective interpolation method (Figs. 2b and d) (Cheng et al., 2017). A spatial decorrelation length scale of 8° was used, which is consistent with a previous gap-filling approach for ocean temperature and salinity (Levitus et al., 2012). A global estimate of salinity $Variance$ in Fig. 2b and Fig. 2d shows that the coastal regions, boundary current systems, and Antarctic Circumpolar Current (ACC) regions have larger $Variance$ than other places, because of either river runoff or mesoscale and sub-mesoscale variabilities. The estimated $Variance$ and the number of data $n$ in each 0.25° and 1-month grid box were combined to perturb the local 0.25°



and 1-month grid mean assuming a Gaussian distribution, which resulted in five perturbed 0.25° and 1-month gridded average fields and inputs into the FFNN approach.

The Monte Carlo dropout approach was employed to estimate the uncertainty due to the FFNN approach. This is one of the most popular ways to estimate machine learning uncertainty, and it does not require any change to the existing model architecture (Stoean et al., 2020; Milanés-Hermosilla et al., 2021). Monte Carlo dropout was operated by randomly dropping some units of the neural network during the model training process. In this way, the approach and parameter uncertainty could be accounted for. In this study, we applied the FFNN to the five above-derived perturbed fields separately to create

five reconstructions each time with the Monte Carlo dropout approach. Then, the spread (i.e., standard deviation) of the $5 \times 5$ = 25 final reconstructions was used to quantify the total uncertainty. With this approach, all major error sources were propagated to the final fields, where the Monte Carlo dropout approach takes into account the reconstruction uncertainty of the FFNN, and the five perturbed gridded fields represent the instrumental and representativeness uncertainty

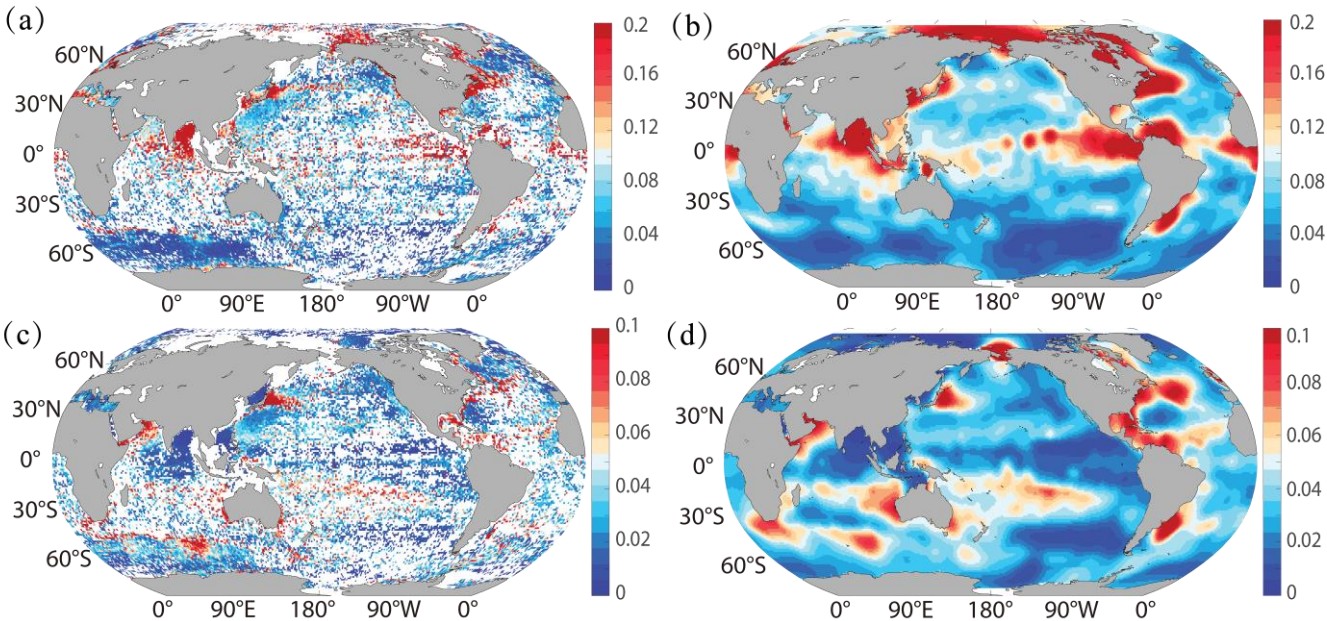

**Figure 2: The estimated *Variance* (a quantification of salinity subgrid variability) of *in situ* observations in this study. The original (left) and objectively analyzed fields (right) are presented for the depths of 20 m (upper panel) and 300 m (lower panel), respectively, as two examples.**

## 3 Reconstruction results

This section presents the reconstruction results and discusses their reliability using multiple examples and thorough

statistical analyses.



### 3.1 Reconstruction of the geographical pattern

We first examine the geographical distribution of the reconstructions for January 2016 as an example. January 2016 was chosen arbitrarily for illustration purposes. Using other times yielded similar results. Figs. 3 and 4 show the salinity anomalies of the new reconstruction, IAP0.25°, in comparison with the IAP1°, ARMOR3D, ADTA, SSTA and *in situ*

observations as of January 2016 at 5 m and 100 m, respectively. It appears that the large-scale pattern of the reconstructed (IAP0.25°) salinity field is closely consistent with the distribution of IAP1° and ARMOR3D, as revealed by *in situ* data. There are significant negative salinity anomalies in the tropical Pacific, Intertropical Convergence Zone (ITCZ), and the Pacific north of the equator, as well as positive salinity anomalies in the Southeast Pacific, Northwest Pacific, and the east coast of Australia. Compared with the very smooth field of IAP1°, the new IAP0.25° salinity field includes more small-scale

signals, indicating greater resolution. Such information has to come from *in situ* data, ADT/SST and USSW/VSSW fields, as they have finer resolution. The previous IAP1° salinity data mainly reveal large-scale patterns of change and there are almost no eddies in the Gulf Stream and the Kuroshio extension regions. The comparison between IAP0.25° and IAP1° reveals a bigger difference in boundary currents and ACC regions with rich eddies.



Figure 3: Spatial distribution of salinity anomalies from IAP0.25°, IAP1°, ARMOR3D, and *in situ* observations at 5 m, as well as the spatial distribution of ADTA and SSTA in January 2016.

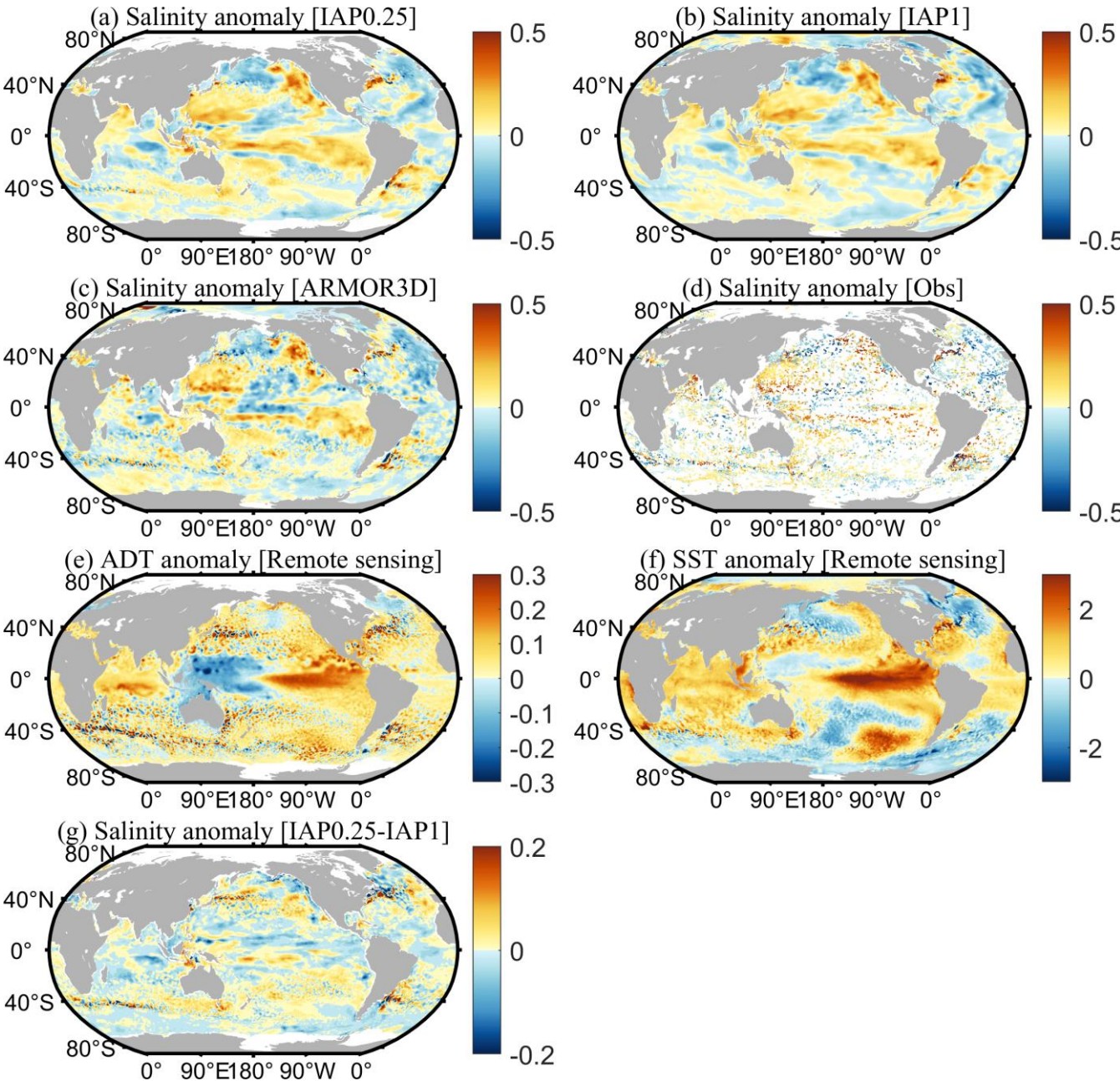

**Figure 4: Spatial distribution of salinity anomalies from IAP0.25°, IAP1°, ARMOR3D, and *in situ* observations at 100 m, as well as the spatial distribution of ADTA and SSTA in January 2016.**

To get a closer look at these critical regions, we concentrate on the salinity reconstruction in the boundary currents regions, e.g., the Gulf Stream (Fig. 5) and the Kuroshio Extension (Fig. 6). Both of these areas are active with mesoscale eddies (Chassignet et al., 2020; Cheng et al., 2014; Frenger et al., 2015; Rhines, 2019), and with very strong net energy growth and net dissipation of mesoscale eddies (Xu et al., 2013).



In the Gulf Stream region (Fig. 5), at a depth of 100 m, IAP1°, IAP0.25°, and ARMOR3D all reveal large-scale positive
salinity anomalies along the North Atlantic coastal regions, negative anomalies in the middle and east, and positive
anomalies extending from southwest (80°W, 15°N) to the northeast (50°W, 30°N). The 300 m salinity shows similar patterns
as that at 100 m, except for larger areas of positive anomalies from southwest to northeast. The similarity among the large-
scale patterns yielded by the datasets suggests the large-scale reconstruction is reliable. However, there are obvious
differences between the four datasets in their ability to describe the fine-scale structures. For example, at 100 m, the
reconstructed (IAP0.25°) field and ARMOR3D show more detail in the salinity structure, especially near the boundary
currents and coastal regions, compared to IAP1°. These anomalies are more consistent with the *in situ* salinity observations
(Figs. 5a–c vs. 5d; Figs. 5e–g vs. 5h).

In the Northwest Pacific Ocean (Fig. 6), IAP1°, IAP0.25°, and ARMOR3D indicate a large area of positive salinity
anomalies in the Northwest Pacific south of 40°N at a depth of 100 m, and negative anomalies north of 40°N. A very smooth
transition zone along ~40°N is apparent for IAP1° (Fig. 6b), but IAP0.25° and ARMOR3D suggest a zone with rich
mesoscale variabilities (Figs. 6a and c).  These findings are more physically plausible because it is the Kuroshio extension
region (Chassignet et al., 2020). Even down to 300 m, there are still rich mesoscale structures seen in the observations within
30°–40°N (Fig. 6h), as represented by both IAP0.25° and ARMOR3D.

A closer look into the two regions suggests that the new IAP0.25° data are capable of resolving the mesoscale signals in the
boundary flow area. However, it also appears that the small-scale anomalies are stronger in the ARMOR3D data than the
IAP0.25° data, which is likely associated with the methodological differences. A comparison between *in situ* observations
and IAP0.25° (Fig. 5a vs. 5c; 5e vs. 5h; 6a vs. 6c; 6e vs. 6h) suggests a closer correspondence of IAP0.25° data with
observations than ARMOR3D, and a more thorough statistical analysis of the errors will be presented in subsequent sections.







**Figure 5: The geographical distribution of salinity anomalies in the Gulf Stream region for IAP1°, IAP0.25°, ARMOR3D, and *in situ* observations in January 2016: (a–d) 100 m; (e–f) 300 m.**



**Figure 6: The geographical distribution of salinity anomalies in the Kuroshio current and its extension region for IAP1°, IAP0.25°, ARMOR3D, and *in situ* observations in January 2016: (a–d) 100 m; (e–f) 300 m.**





## 3.2 Reconstruction of the vertical structure

In addition to the spatial distribution at a given layer, it is desirable to investigate the vertical structure of the salinity field (i.e., to what extent the vertical structure can be restored by the new approach). This section seeks to address this question with two examples: in the Gulf Stream region (along 35.375°N), and in the Kuroshio extension region (along 35.375°N) (Figs. 7 and 8).

Along the Gulf Stream section, satellite remote sensing SSS data (Fig. 7a), independent from our analysis, suggest a general negative salinity anomaly associated with substantial variability. This reveals the existence of both large-scale change and mesoscale variability. It can be seen from Figs. 7b and c that both the IAP0.25° and IAP1° anomalies are generally negative near the sea surface, indicating the near-surface signals can be restored with IAP0.25°. The smoothness of the IAP1° data is associated with its mapping method (Cheng and Zhu, 2016), which focuses more on the large-scale patterns and effectively filters out small-scale variations. Moreover, the small-scale, along-section fluctuations revealed by the *in situ* profile data (Fig. 7c) are well reflected in the reconstructed (IAP0.25°) field (Fig. 7d), but not in the IAP1° data (Fig. 7b), highlighting that IAP0.25° is capable of correctly reflecting the small-scale features. ADT data reflect the integrated effect or thermosteric (linked to temperature) and halosteric (linked to salinity) effects over the full volume (Llovel and Lee, 2015; Wang et al., 2017). To the first order, in the thermocline regions, the ADT change should correspond better with subsurface salinity, which is supported by Fig. 7 (red line in Fig. 7a vs. 7d); for example, the large positive anomaly near 48°W, 60°W, and 68°W revealed by both the *in situ* salinity profile and ADT data. This indicates a positive contribution of ADT to the reconstruction.

Along the Kuroshio extension section (Fig. 8), the SST and ADT are more consistent with each other than in the Gulf Stream (Fig. 7). There is a deep mixed layer (deeper than 500 m) in this region, so ADT is more influenced by mixed-layer dynamics. In this case, both ADT and SST could support the high-resolution salinity reconstruction because mixed-layer dynamics are associated with strong air–sea heat and freshwater exchanges. For example, the high ADT, SST, and salinity anomalies at 155°–160°E, around 180°E, and negative anomalies at 178°E and other places in the upper 500 m are all consistent (Figs. 7a, c, d).

In summary, the investigation of these two specific cases increases the degree of confidence in the reconstruction. It seems that certain mesoscale structures can be realistically restored.



**Figure 7: The distribution of ADTA, SSTA, and SSSA observed by SMAP on the sea surface, (ADTA and SSTA share the right-hand *y*-axis), as well as the vertical section along 35.375°N of the observational *in situ* salinity anomalies, subsurface salinity anomalies from IAP1°, and IAP0.25° gridded data in the Gulf Stream region.**



**Figure 8: As in Fig. 7, but for the Kuroshio current and its extension region along 35.375°N in January 2016.**

### 3.3 Overall reconstruction performance

In the previous two sections, examples were presented to illustrate the reconstruction performance. However, a more thorough statistical examination is still required, which is provided in this section. To quantify the overall error compared



with observations, we calculated the RMSE between IAP0.25°, IAP1°, ARMOR3D, EN4, and the observation data, respectively (denoted as IAP0.25_RMSE, IAP1_RMSE, ARM_RMSE, and EN4_RMSE), and the results are presented in Figs. 9a–d at different layers from the surface down to 2000 m over the globe and in three major basin regions during 1993–2018. Figs 9e–h show the differences between IAP1_RMSE, ARM_RMSE, EN4_RMSE, and IAP0.25_RMSE. First, it seems that IAP0.25_RMSE is smaller than all other data products on both a global- and basin-average basis. In the global

area, the maximum values are 0.37 psu for IAP0.25_RMSE, 0.50 psu for IAP1_RMSE, 0.55 psu for ARM_RMSE, and 0.56 psu for EN4_RMSE near the sea surface. Globally, although a reduction in RMSE can be seen from the surface down to at least 1000 m, the upper 0–200 m shows a more significant reduction for IAP0.25_RMSE compared with other data, consistent with more mesoscale variability in the upper ocean. The global 0–2000 m averaged global reduction in RMSE for IAP0.25_RMSE is 0.0164 psu, 0.0194 psu, and 0.0208 psu compared to IAP1_RMSE, ARM_RMSE, and EN4_RMSE,

respectively (Table 2). The smallest RMSE is apparent for IAP0.25° at all 41 vertical levels over the globe and in the three major basins, suggesting that IAP0.25° best represents the *in situ* observations.

Besides, the reduction in IAP0.25_RMSE is more remarkable in the Atlantic Ocean than in the Pacific and Indian oceans. A previous dataset intercomparison study suggested that the Atlantic Ocean shows larger spread among different data products, even with more observations (Frederikse et al., 2018). This larger error reduction in the Atlantic basin for IAP0.25° suggests

that signals smaller than 1° might play an important role for a reliable reconstruction.

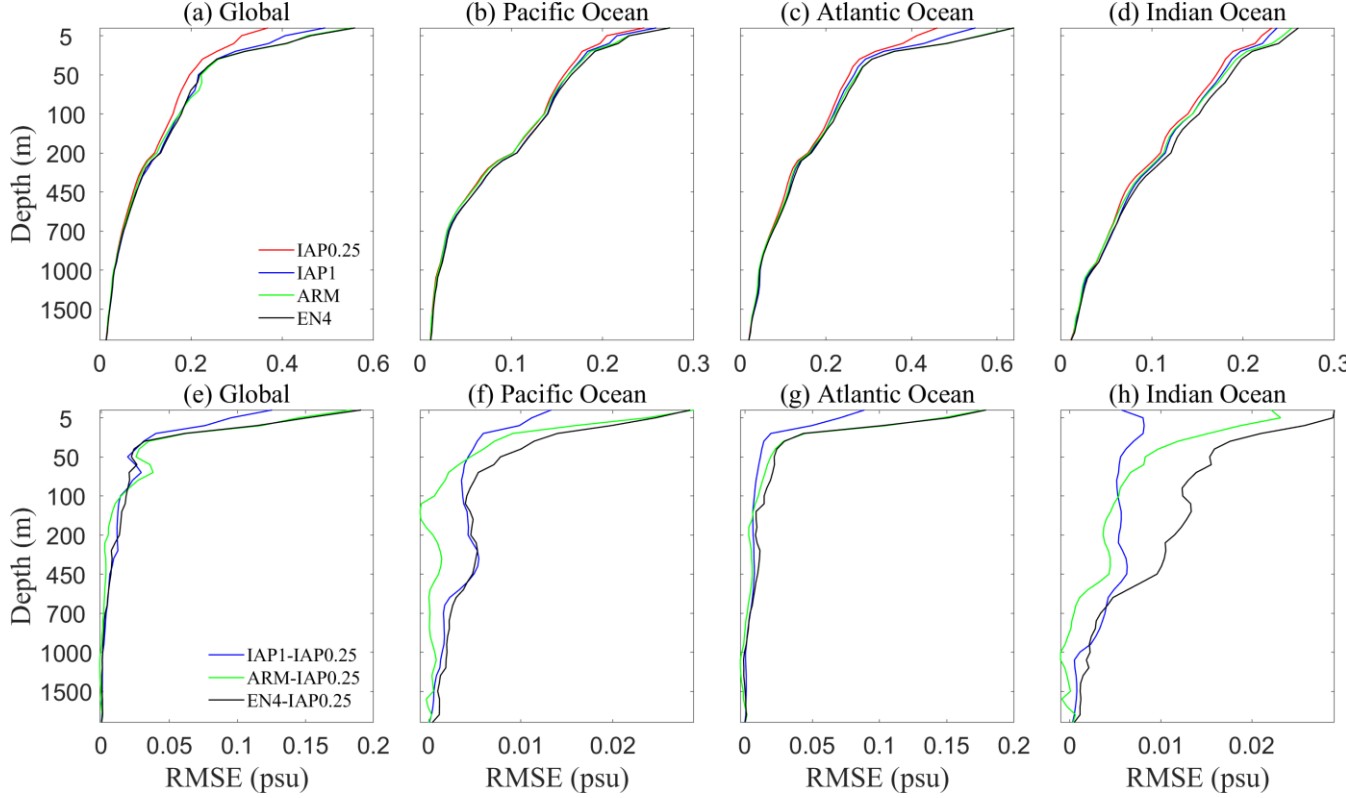



**Figure 9: (a–d) Vertical distribution of RMSE for IAP1°, ARMOR3D, EN4, and IAP0.25° for the globe and three major basin regions during 1993–2018. (e–h) Vertical distribution of RMSE for IAP1°, ARMOR3D, and EN4 minus IAP0.25°, respectively.**

**Table 1. RMSE of 0–2000-m mean salinity for the IAP1°, ARMOR3D, EN4, and IAP0.25° datasets (unit: psu).**

| Region | IAP1_RMSE-IAP0.25_RMSE | ARM_RMSE-IAP0.25_RMSE | EN4_RMSE-IAP0.25_RMSE |
|---|---|---|---|
| Global | 0.0164 | 0.0194 | 0.0208 |
| Pacific | 0.0036 | 0.0028 | 0.0056 |
| Atlantic | 0.0101 | 0.0161 | 0.0190 |
| Indian | 0.0042 | 0.0046 | 0.0094 |

Furthermore, to identify the causes of the most significant errors, the spatial distribution of RMSE for IAP0.25° and IAP1° minus IAP0.25° for three representative layers and 0–2000-m averages are presented in Fig. 10. Compared with IAP1°, the reduction in RMSE for the IAP0.25° data is more dramatic in the western boundary currents, ACC region, and coastal

regions (Figs. 10e–h). However, the RMSE for IAP0.25° remains larger in the western boundary currents, such as the Kuroshio, Gulf Stream, and Brazil Current regions, as well as the ACC regions, compared to the open ocean. This finding is consistent with stronger mesoscale variations in these regions in the upper ocean (Frenger et al., 2015; Chassignet et al., 2020; Rhines, 2019) (Figs. 10a–d). In particular, at 5 m, IAP0.25_RMSE is larger in the Bay of Bengal and Arabian Sea, mainly because, in these areas, salinity changes are strongly affected by river runoff, surface fluxes, and ocean currents

(Skliris et al., 2014; Adler et al., 2018; Liu et al., 2022). Larger errors in the ITCZ regions are also apparent at 5 m and 100 m, corresponding to strong surface freshwater fluxes (Liu et al., 2022).



Figure 10: Geographical distribution of RMSE: (a–d) IAP0.25_RMSE; (e–h) IAP1_RMSE minus IAP0.25_RMSE.

The temporal variation of the global 0–2000-m mean RMSE from 1993 to 2018 is presented in Fig. 11 for both IAP1° and

IAP0.25°, to show the change in reconstruction performance with time. First, it appears that the error reduces with time,

especially since the early 2000s, probably in association with there being more salinity observations available from the Argo

network (Yan et al., 2021; Chen et al., 2018; Wong et al., 2020; Roemmich et al., 2019), which helps to increase the

reconstruction accuracy. The RMSE has been increasing slightly in recent years, probably because of the inclusion of more

real-time Argo data (with lower data quality) in our analyses. Nevertheless, IAP0.25_RMSE is smaller than IAP1_RMSE for

the global ocean, by ~11% on average. The reduction is about 0.013 psu. The biggest reduction in RMSE is seen in the

Atlantic Ocean (by ~0.025 psu, ~17%), confirming our results in previous sections. It is interesting that there is a strong

seasonal fluctuation of RMSE: it is larger in the Northern (Southern) Hemisphere summer (winter), likely because of there

Earth System
Science
Data

Open Access / Discussions

being less data in the Southern Ocean owing to the sea-ice coverage and severe weather (Zweng et al., 2019; Gould et al.,

2013; Auger et al., 2021; Durack, 2015).

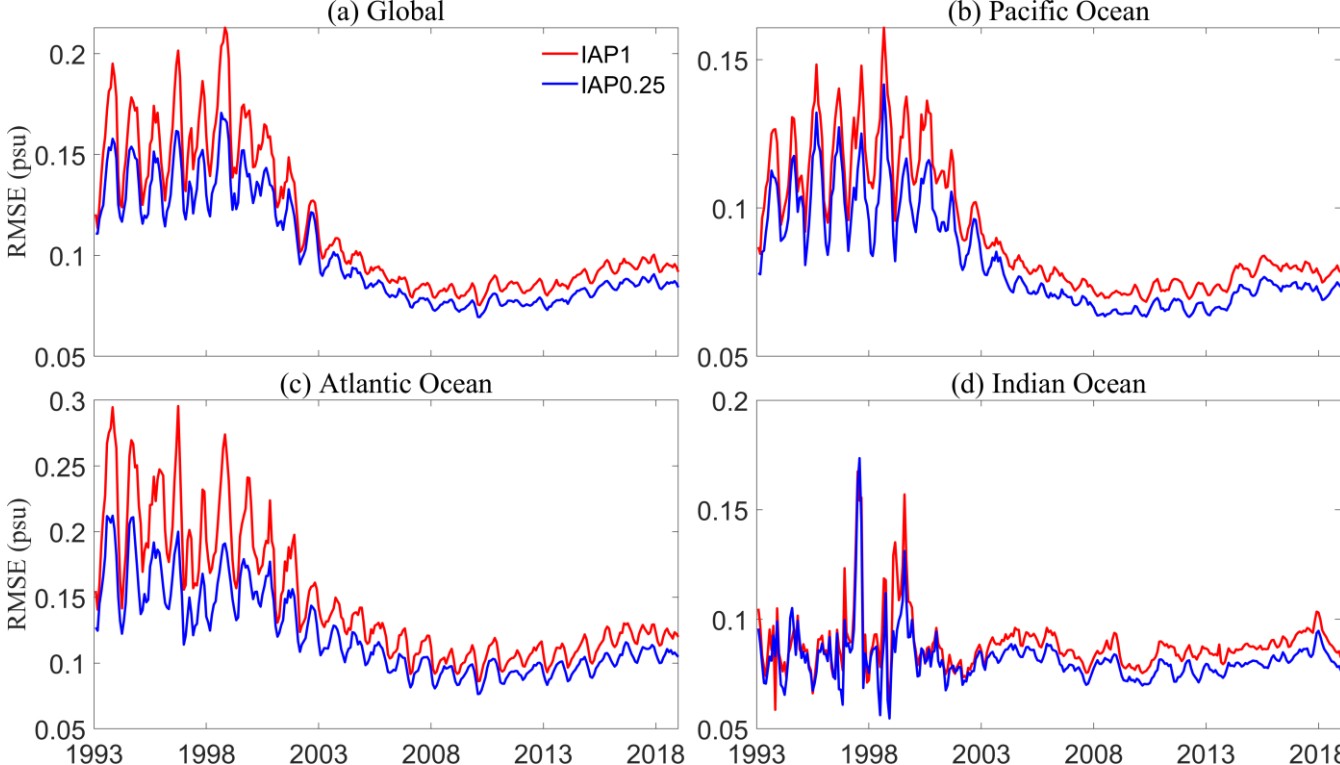

**Figure 11: The 0–2000-m average RMSE time series for the IAP0.25° and IAP1° datasets for the globe and three ocean basins from 1993 to 2018.**

In summary, statistical analysis suggests a small RMSE between the reconstruction field of IAP0.25° and the *in situ* salinity

observations, indicating an improved performance of the high-resolution reconstruction (IAP0.25°) in this study compared

with other products (IAP1°, ARMOR3D, and EN4). However, it should be noted here that the observations were processed

by our research group and used to train the FFNN model, meaning the observations are not independent of the final

reconstruction. Thus, besides the analyses reported so far, an independent validation was warranted, the results of which we

report in the next section.

**4 Five-fold cross validation and uncertainty estimate**

In this section, we use the results from a five-fold cross validation to further evaluate the new reconstruction with the

withheld independent observations. For this test, we trained the FFNN model with the training set (80% of the full set) and

use the withheld 20% of data to evaluate the performance. This process was repeated five times, so the full set could be used



for evaluation (i.e., the full set was divided into five sets and each time one set was used as the testing data and the other four

sets were training sets).

Fig. 12 shows the density distribution between the reconstructed salinity value and the *in situ* observations for the training set (a, c) and testing set (b, d) at the depths of 10 m and 100 m, separately. The diagonal line denotes a perfect reconstruction, i.e., the reconstruction perfectly matches the observations. Any errors in observations and/or reconstructions can lead to scattering of the distribution. It appears that the highest density locations are along the diagonal line and most of the

reconstruction–observation differences are within ±0.2 psu for both the training and testing set, indicating that the reconstructions are not obvious biased. The density distribution for the testing set is not visibly different from the training set, suggesting that our FFNN model does not overfit the data in the training set, meaning the reconstruction method is robust in the independent data.

Besides, the RMSEs between the reconstructed salinity field and the *in situ* observations of the independent testing data were

calculated (Fig. 12e). The difference in RMSE between the training set and testing set is a measure of the model "degradation", which quantifies the generalization of the model. The RMSE of the testing set is only marginally higher than that of the training set: at 10 m, it is 0.2613 psu for the training set and 0.2692 psu for the testing set; and at 100 m, it is 0.1402 for the training set and 0.1403 for the testing set. Meanwhile, the correlation coefficient is only slightly lower for the testing set: at 10 m, the correlation coefficient decreases from 0.7068 (training set) to 0.6857 (testing set); at 100 m, it

decreases from 0.6245 to 0.6233. Furthermore, the averaged RMSE for each depth from the surface to 2000 m was calculated for the training and the testing set, respectively (Fig. 12e). It appears that the two RMSE curves are very close to each other at all depths. The degradation percentage of RMSE was then calculated to better illustrate the difference: the difference in RMSE between the testing and training set was divided by the RMSE of the training set (Fig. 12e). The RMSE degraded by 5.49% in the surface layer, but the degradation rate decreased to less than 0.5% from 30 m. Since the testing set

is composed of completely independent data, this test indicates that the FFNN model does not experience serious overfitting in our salinity reconstruction, and the method is valid.



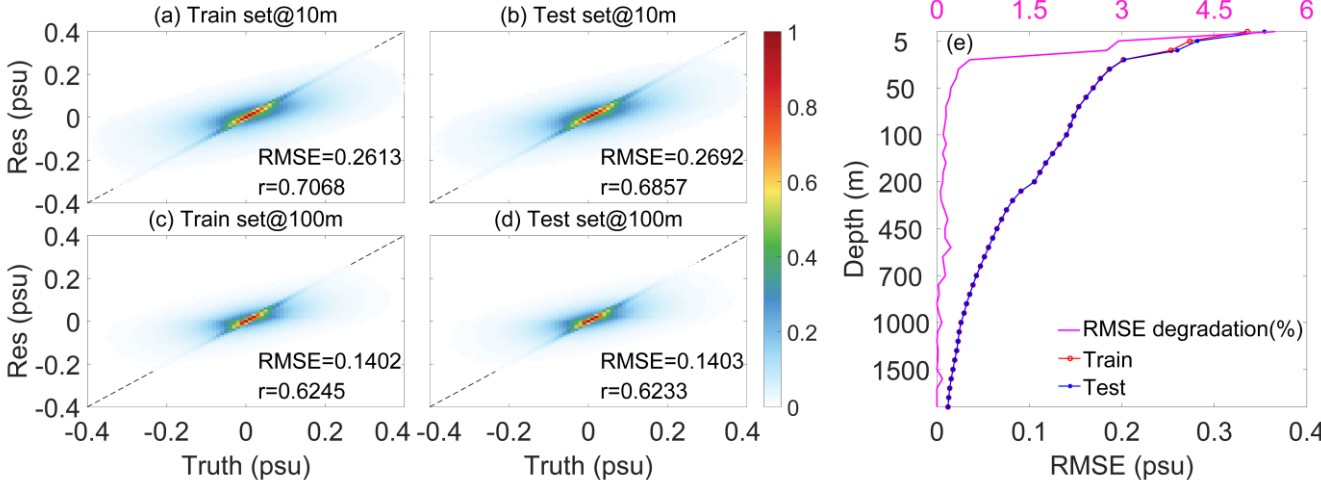

**Figure 12: (a–d) Density distribution diagrams at depths of 10 m and 100 m for the training set and testing set. (e) Vertical distribution of RMSE for the training set and testing set for global region (bottom *x*-axis) and the RMSE degradation from the training set to the testing set (top *x*-axis). The correlation coefficient is denoted by "r", and the color-coded blocks represent the density of samples.**

## 5 Evaluation of the major climatic patterns

It was shown in the previous section that IAP0.25° is capable of reconstructing salinity with more mesoscale signals. However, it remains to be quantified whether the large-scale pattern of salinity change associated with climate change can be well represented in the new reconstruction. Here, we examine the long-term trends in ocean salinity using the Salinity Contrast (SC) index proposed by Cheng et al. (2020). The SC is defined as the difference between the salinity averaged over high-salinity regions ($V_{High}$, where salinity is higher than a climatological global median, $S_{clim}$) and low-salinity regions ($V_{Low}$, where salinity is below $S_{clim}$). It was calculated each month over the three-dimensional ($x$, $y$, $z$) ocean salinity field:

$$SC(t) = \frac{\iiint_{V_{High}} S(x,y,z,t)\,dV}{\iiint_{V_{High}} dV} - \frac{\iiint_{V_{Low}} S(x,y,z,t)\,dV}{\iiint_{V_{Low}} dV}, \tag{1}$$

where $x$, $y$, and $z$ are the three dimensions of latitude, longitude, and depth, respectively. The terms $V_{High}$ and $V_{Low}$ were both determined on the basis of the climatological salinity field.

The SC at the sea surface is called SC0; and for the 0–2000 m volume, the SC is termed SC2000. It can be seen from Fig. 13 that the IAP0.25° and IAP1° datasets have consistent long-term changes of SC. The global-scale SC2000 increases significantly from 1993 to 2018, with a very significant linear trend. The increase of this index indicates that the phenomenon of "fresh gets fresher, salty gets saltier" is more obvious, which is mainly driven by the changes of "wet get wetter, dry get drier" in the global water cycle (Cheng et al., 2020; Marvel et al., 2017; Skliris et al., 2014).



**Figure 13: Salinity contrast time series for IAP1° and IAP0.25° data in global regions from 1960 to 2020: (a) at the surface; and (b) in the upper 2000 m.**

Besides the global-scale SC metric, we also present the salinity time series over the globe and in different ocean basins for

IAP1° and IAP0.25° from 1993 to 2018 for both the SSS and 0–2000-m averaged salinity (S2000), in Figs. 14 and 15. We

divided the oceans into the Pacific Ocean (35°S to 60°N), Atlantic Ocean (35°S to 75°N), Southern Ocean (70°S to 35°S),



North Indian Ocean (0° to 30°N), and South Indian Ocean (35°S to 0°). The uncertainty was quantified by the approach described in section 2.3.3, where the ±2 standard deviation error range is provided, which corresponds to a 95% confidence
level. The two datasets continue to show high consistency:

Globally, and in the major ocean basins, IAP0.25° and IAP1° are consistent for SSS (Fig. 14). The SSS of both IAP1° and IAP0.25° shows an upward but statistically insignificant trend from 1993 to 2018. The SSS increases steadily in the Atlantic Ocean (35°S–75°N) and Southern Ocean (70°S–35°S) (Figs. 14c and d). The Pacific SSS indicates a strong decadal-scale fluctuation. In both the North and South Indian Ocean, the SSS exhibits a weak long-term decreasing trend (Figs. 14e and f).

The global mean S2000 time series (Fig. 15a) shows an increasing trend between 1993 and 2018, consistent with previous studies (Ponte et al., 2021). With more freshwater input into the ocean, the ocean salinity is expected to decrease, and thus this increase of S2000 indicates that the impact of terrestrial ice-melt is difficult to resolve from salinity observations. The data drift in salinity observations is most likely responsible for the observed increase, but this issue has not been resolved yet (https://argo.ucsd.edu/faq). S2000 declines steadily in the Pacific basin, but increases sharply in the Atlantic basin from 1990

(Figs. 15b and c). The increased SC between the Pacific and Atlantic oceans is due to increased inter-basin transport of water vapor from the Atlantic to the Pacific (Reagan et al., 2018; Curry et al., 2003). The decadal fluctuations of S2000 in the Atlantic are greater than those in the Pacific, especially in the North Atlantic, and are in-phase with the Atlantic Multidecadal Oscillation (Skliris et al., 2020; Reverdin et al., 2019). S2000 increases in the North Indian Ocean (Fig. 15e) and decreases in the South Indian Ocean (Fig. 15f), mainly because the climatological salinity is high in the former and low

in the latter.

In summary, through our investigation of the global and basin salinity time series, we have been able to further confirm that IAP0.25° can capture the integrated property of salinity changes compared with IAP1°. The estimated 95% confidence intervals are generally consistent for different basins, which is about ±0.01 psu for SSS and ±0.002 psu for the 0–2000-m average. Although the error range of this new estimate is larger than IAP1°, because more sources of uncertainty are

accounted for (IAP1° mainly accounts for mapping and instrumental error), this new estimate could still be an underestimate because of the neglect of systematic systematically biases (which are currently poorly known) in this study.

**Figure 14: The global (a) and basinal (b-f) averaged SSS time series from 1993 to 2018 for IAP1° and IAP0.25° respectively. Both monthly and 12-month running smoothed time series are presented, all data are relative to a 1993–2015 baseline.**

Earth System
Science
Data


**Figure 15: The global (a) and basinal (b-f) 0-2000m averaged salinity time series from 1993 to 2018 for IAP1° and IAP0.25° respectively. Both monthly and 12-month running smoothed time series are presented, all data are relative to a 1993–2015 baseline.**

## 6 Data availability

The code mainly used in this paper includes data processing, optimal model building and the result prediction. The
reconstructed IAP0.25° dataset is available at http://dx.doi.org/10.12157/IOCAS.20220711.001 (Tian et al., 2022) or
http://www.ocean.iap.ac.cn/ or http://www.ocean.iap.ac.cn/ftp/cheng/IAP_v0_Ocean_Salinity_0p25_FFNN_0_2000m/.
Here we provide global ocean salinity gridded product at 0.25° × 0.25° horizontal resolution on 41 vertical levels from 1-
2000m, and monthly resolution from 1993 to 2018.





## 7.Summary and Discussion

This study used an FFNN approach to reconstruct a high-resolution (0.25° × 0.25°) ocean subsurface salinity dataset (1–2000 m) for the period 1993–2020, in which the spatial and temporal information (time, longitude, latitude, depth), and satellite remote sensing data (ADT, SST, SSW) were input into the reconstruction. By training the functional relationship between input variables and truth values (observed salinity), the reconstruction model was established. For the IAP0.25° salinity data, the global and regional reconstruction performance was evaluated using several different tests. In brief, we show that: (1)

IAP0.25° salinity data maintain the large-scale information from IAP1° gridded data, and also include some smaller-scale (i.e., mesoscale) information from satellite remote sensing information. It thus serves as an improvement on the currently available IAP data. It has been shown that IAP1° performs best for large scale patterns and long-term climate change and variabilities (Cheng et al., 2020; Reed et al., 2022; Sohail et al., 2022) compared with many available gridded products, thus this continuity is an advantage of the new IAP0.25° product. (2) The machine learning approach shows efficiency in merging

different kinds of Earth observations, providing a useful addition to the available data assimilation and objective analysis methods. (3) We used a five-fold cross-validation to evaluate the reconstruction performance with withheld independent observations, showing that the method is robust and can be reliably used for salinity reconstruction.

Although the validity and advantages of the FFNN approach have been demonstrated, there are some caveats and limitations to this approach, related to which we provide some future guidance. This study used a probabilistic approach to quantify the

uncertainty, which is a practical strategy, and with this approach all major error sources are propagated to the final fields. However, uncertainty assessment with machine learning is still a challenge for many available products, and this issue requires future analysis: i.e., how sensitive the FFNN method is for parameter choices such as the number of hidden layers and neurons.

Compared with the ARMOR3D data, it seems that the small-scale signals are weaker in IAP0.25° (for example Fig. 5 and

Fig. 6), indicating a difference of efficiency with which signals of the remote sensing data are transmitted into the salinity reconstruction. Therefore, an intriguing scientific question is what is the key factor determining the efficiency? Addressing this question requires a dedicated study that incorporates an intercomparison among other approaches.

As a coarse-resolution product, IAP1° is important in high-resolution reconstructions and provides a critical large-scale precondition. The uncertainty (or biases) in this coarse-resolution field will propagate into the reconstruction, but how the

uncertainty propagates and contributes to the final estimate are still open questions. One of the aims of this study was to ensure the continuity from IAP1° to IAP0.25° data, because work has already shown superiority of IAP1° in large-scale reconstructions compared with many available datasets (Cheng et al., 2020). Thus, this study further advances the prior findings of Cheng et al. (2020). Further experiments are needed to quantify the sensitivity of the results to this preconditioning.






**Author contributions.** TT developed the related algorithm, generated and evaluated the IAP0.25° dataset, and wrote the paper. LC conceptualized and supervised this study. GW and HL participated in data processing and sample selection. JA, JZ and JS edited and revised the paper.

**Competing interests.** The contact author has declared that none of the authors has any competing interests.

**Acknowledgments.** Many thanks are extended to the institutions or organizations that provided the data used in this paper (the specific datasets and sources are as described in the text). We also thank all anonymous reviewers for their detailed and constructive comments.

**Financial support.** This study was supported by the Strategic Priority Research Program of the Chinese Academy of Sciences [grant number XDB42040402] and the National Natural Science Foundation of China [grant numbers 42122046
and 42076202].

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
