# Peer review of "Reconstructing ocean subsurface salinity at high resolution using a machine learning approach"

_Earth System Science Data, 2022_

## Referee Comment (RC2)

**Object**: Referee report for Earth System Science Data
**Paper**: Reconstructing ocean subsurface salinity at high resolution using a
machine learning approach
**Authors**: Tian Tian & al.

▪·▪·▪·▪·▪·▪·▪·▪·▪·▪·▪·▪·▪·▪·▪·▪·▪·▪·▪·▪·▪·▪·▪·▪·▪·▪·▪·▪·▪·▪·▪·▪·▪·▪·▪·▪·▪·▪·▪·▪·▪·▪·▪

In this work the authors describe a new product of global subsurface salinity at a high resolution (0.25°x0.25°) covering 41 vertical levels (in the range 1-2000m), named IAP0.25°.
This product is obtained using a Feed Forward Neural Network model designed by the authors, which is trained from several input variables taken from satellite sets and reanalysis to reconstruct the subsurface salinity product. Interpolated *in situ* observations of salinity profiles has been considered as ground truth to compare reconstruction findings. Finally IAP0.25° is evaluated by comparison with three independent ocean products of salinity.

The overall presentation of the new product, results and evaluation metrics is of high quality, hence in my opinion the manuscript should be considered positively for publication in ESSD. I would suggest a minor revision, mainly because I find that improvements in presentation and some additional comments have to be considered. I give also line by line suggestions to ease revision.

- The paper is quite long: I find it is possible to shorten some parts by avoiding repetitive sentences or being more concise and direct in some subsections. This is particularly true, for example, for the Introduction. I believe that shortening the manuscript would aid the overall readability, hence facilitating the choice of using the authors' dataset.

- The link of the IAP0.25° DOI should be given for the English version of the website page. Indeed you end up in the Chinese one and when you switch to English you are brought back to the home page, which is confusing. In the website the dataset is told to cover 0-2000m instead of 1-2000m (which is correctly reported in the manuscript and seen from the netcdf downloaded)

- The FFNN model description needs some additional details. In general the authors have fully described data and reconstruction and evaluation, but less attention has been paid in motivating the NN choice and structure. I believe that adding this kind of comments would aid in understanding the background ratio.

- Understanding which input mostly influences the salinity reconstruction and causes greater propagation error would be very interesting in this work, and in my opinion would enhance completeness of the presentation (this is stated as a future step in the Summary section, hence to be considered only as a suggestion).

- In several points some statements sound very vague or too general (see line by line comments hereafter). I suggest to be more precise. For example the authors could revise how they refer to machine learning in a vague way: it might be more interesting to focus on neural networks only, since this is the model choice for this study.

▪·▪·▪·▪·▪·▪·▪·▪·▪·▪·▪·▪·▪·▪·▪·▪·▪·▪·▪·▪·▪·▪·▪·▪·▪·▪·▪·▪·▪·▪·▪·▪·▪·▪·▪·▪·▪·▪·▪·▪·▪·▪·▪

1. Introduction
- (54) typo *below the ocean surface
- (62) what to you mean with sufficient data quality? This is vague
- (75-76) state better the limitations
- (77-78) in this study you inspect NN for reconstruction of salinity field only, not for other ocean subsurface fields. state better
- (86) I would not talk about "major deficiencies", which sounds too strong for the subsequent comments. Maybe *some differences, *some limitations or similar
- (94-96) a bit repetitive with (77-79). revise by shortening

- (97) the second objective introduced does not sound as an objective
- (103) currently your section 6 is of Data availability (this could become your last section 7, probably more appropriate)

2. Data & Methods
- (107-109) this should be said in the introduction, not here. I would not say "a machine learning model", which sounds very general; instead point out it is your model- this happens also in other points of the paper.
- (110, 115, 121) sentences "data were from.." could be improved with expressions as: we use, we downloaded, data was extracted from...
- (112) sounds like you have done the optimal interpolation
- (116) take off "and extrapolating"
- (119-120) take off last sentence, you say this at the end of the 2.1.1 ssec, repetitive
- Table 1. Should be better organised, I suggest you should have: Data type, Variable, Dataset, Data Source, Horizontal resolution, Vertical coverage and resolution, Time period, Reference, DOI; hence correct information given (e.g. for SST you would have Variable:SST, Dataset: OISST, Data Source: NOAA, etc.). Pay attention to classifying Salinity observations as Input, since indeed you use it as ground truth to which you compare the model output.
- Eventually insert DOIs in text for each product for completeness
- (131) say something more of interpolation technique adopted
- (139-140) state better, e.g. anomaly profiles are derived by subtracting monthly climatologies from salinity profiles.
- (140-141) not clear to me, to state better
 - why do you say that SSA were averaged in 0.25 resolution? Did you mean interpolated?
- (146) "certain tests" is too vague. Say if you tested your method on raw profiles and found no differences, or say something more about the tests (without needing to show figures or results about)
- (150) remove "including", since you are stating all sets
- (156-157) remove "which have spatial resolution of 0.25", repetitive
- (169) You should start this subsec by describing the FFNN, not the data. These first sentences could be moved in data subsec or later on.
- (186-187) complexity of a NN depends not only on how many neurons or layers are chosen, but also on type of layers and activation functions.
- (190-192) make it shorter and less repetitive
- (195) you should not refer to your model generically as "a FFNN" but "the FFNN"
- (196) take off "based on training set" uninformative
- (199) I would not talk about "reconstruction effect"
- Figure 1. when showing the output information you give several maps of subsurface salinity anomalies, since your method is reconstructing each time step separately from the others I suggest you leave only one map also in the figure. Caption: I suggest you change in "Schematics of the FFNN architecture for subsurface salinity anomalies reconstruction (IAP0.25)"
- which is your cost function minimised for training? did you use a GD algorithm? say more
- (209) I would mention that the independent dataset are those introduced in ssec 2.2
MANCA 2.3.3
- (243-244) is variance the standard deviation? usually variance=std**2
- (256) I would add "estimate machine learning methods uncertainty"
- (263) missing ending dot
- Figure 2. (b&d) it seems to me that there might have been higher values than 0.2, flattened to be exactly 0.2. if this is the case please adapt colorbar with pointed end for out-of-range values. Eventually it would be interesting to have the four panels shown with same colorbar limits to be able to compare effectively two different depths' results (if graphically informative)

3. Reconstruction Results
- (275) typo: take of "as"
- (276) you should quantify the consistency between your reconstruction and IAP1 & ARMOR3D with some metric

- (280) take off "indicating greater resolution", uninformative. Next sentence take off "of change", the patterns you are showing are of one particular month.
- (282) How did you perform subtraction of IAP0.25 and IAP1? Did you first have IAP1 interpolated?
- I find it curious that you first show results of 5m and 100m depth globally, but then you turn to 100m & 300m for specific regions. Why is this? if there is no reason I would try to be consistent between global and detail analysis
- Figure 3 & 4. Refer in caption to each product by giving subfigures letters. You don't mention comparison between IAP025 and IAP1 in the caption (panel e that could be given close to the two products images)
- (298) You talk about reliability of the reconstruction by giving very qualitative comments, this should be quantified for robustness via some metric
- (327) typo: Figure 2b and *d
- (334) It is not clear to me when you state that ADT change should correspond better with subsurface salinity. I would suggest to state the whole ADT paragraph better
- (345) take off "certain", it sounds vague
- Figure 7a. Legend should be put all together possibly internal to the image. I suggest to put longitude coordinates on the top of the figure to aid readability of comments.
- (359) "seems that" does not sound as a statement. Substitute "in the global area" with "At a global scale"
- (363-365) very heavy to read, try to improve
- Figure 9. The best way for comparing graphically rmse over these regions is to have same figure limits for the upper panels (same for the lower panels).
- (395) take off parenthesis. why did you include lower quality data?
- (397) I would say even something about lowest reduction
- Figure 11. The y-axis limits should be the same for all subfigures for a more informative graphical comparison of results. In the caption I would mention that time series are averaged over the different areas
- (404) *reveals, instead of suggests, more precise
- (406) *with the other products considered

4. Five-fold cross validation and uncertainty estimate
- (412) I would recall these independent observations, referring also to subsec 2.2.
- (421) typo: * are obviously not biased
- (424-435) it is confusing how you comment the findings. First you talk about Figure 12e, then 12a-12d and then 12e again. Revise the whole paragraph, possibly shortening
Figure 12. in the caption the sentence on correlation coefficient should be found when talking about panels a-d

5. Evaluation of the major climatic patterns
- (454) saying a "very significant linear trend" is too vague. Have you computed it with a p-value? what is the annual rate of change? state better
- Figure 13. the title shoud be Salinity Contrast index at Surface/0-2000m. I believe no y label is needed, and anyways should not be salinity since SC is shown
- (461) *for both the salinity at surface and averaged over the 0-2000m volume
- (475) To me the sentence regarding increased SC is not clear
- (479) you re talking about anomalies, why should average values impact the change? state better
- (486) typo: *systematic twice
- Figure 14 & 15. if possible all subfigures should have same y axis limits to ease comparison. Are these salinity anomalies?

6. Data availability
- (494) take off *mainly
- (498) *at a monthly resolution

7. Summary and Discussion
- (502) *were given as inputs to the FFNN algorithm for reconstruction
- (508) it is vague to say that IAP025 performs best compared with many available gridded products. state better
- (510)  you should mention that for point (2) you are referring to subsurface salinity field. Indeed I don't see why separating this point from previous one. Instead I would say something of your findings of rmse or climatic patterns, which are not cited in this summary
- I suggest that this section should be revised trying to make it more appealing, and pointing more clearly to your findings.

---

## Author Comment (AC1)

**We appreciate the community's comment from Dr. Etienne Pauthenet. Please find below a point-by-point response to comment. The comments are in black, our response is in blue, text changes are in orange.**

Manuscript number: essd-2022-236

MS type: Data description paper

Title: Reconstructing ocean subsurface salinity at high resolution using a machine learning approach

Author(s): Tian Tian, Lijing Cheng, Gongjie Wang, John Abraham, Shihe Ren, Jiang Zhu, Junqiang Song, and Hongze Leng

Submission date for revisions: 4th Oct 2022

CC #1:

I would like to thank the authors for the very interesting work and the thorough validation of the results. However I believe there is overstatements in the presentation of the paper compared to what the authors actually did, that would require some clarification before publication. The paper is presented as a reconstruction of gridded salinity fields from observation, but it is in fact an upscaling of the IAP1 product from 1 degree to 0.25 degree resolution. I have three main comments.

**Re:** We'd like also thank Dr. Etienne Pauthenet for the insightful comments and introduction of their newly published paper (a regional reconstruction in Gulf Stream with machine learning approach). In response, we have clarified several issues and cite/discuss the suggested paper from Dr. Etienne Pauthenet in the revised manuscript. These comments help to improve our manuscript. However, we don't think there is "overstatement" in the presentation of our paper by doing gridded averages, as discussed below (it is a common practice to do the gridded average on observations before mapping/spatial interpolation).

1 - This paper uses data already gridded by Global Circulation Models (GCM) as inputs (IAP1 and SSW). The IAP1 data is also "interpolated into unified monthly and 0.25°× 0.25°spatial resolution fields" (L131-L132). Therefore the following sentences are overstatements because they are not mentioning the presence of GCM in the inputs already gridded at 0.25x0.25 resolution :

—> L94-L96  : "This paper explores the possibility of using a machine learning approach in which in situ observations and remote sensing data are merged to reconstruct the salinity changes at a 0.25°×0.25°horizontal resolution and a monthly temporal resolution from the surface down to a depth of 2000 m."

**Re:** Several sentences have been added to expand our introduction to IAP1° product, which is an already well-established dataset. "This product combines *in situ* salinity profiles with coupled model simulations (from phase 5 of the Coupled Model Intercomparison Project ) to

derive an objective analysis with the Ensemble Optimal Interpolation approach (Cheng et al., 2020)".

—> L500-L503 : "This study used an FFNN approach to reconstruct a high-resolution (0.25°×0.25°) ocean subsurface salinity dataset (1–2000 m) for the period 1993–2020, in which the spatial and temporal information (time, longitude, latitude, depth), and satellite remote sensing data (ADT, SST, SSW) were input into the reconstruction. By training the functional relationship between input variables and truth values (observed salinity), the reconstruction model was established."

To correct this would require a change of title, it is not a "reconstruction" but an "upscaling". And also correct the abstract, introduction, and conclusion, by stating first that the network is taking IAP1 as an input. I believe IAP1 is the main source of information the network is learning from. An analysis of the relative importance of each input for each output could answer this question (see figure 10 and 11 of Pauthenet et al 2022).

**Re:** First, the above-mentioned sentence has been revised to "This study used an FFNN approach to reconstruct a high-resolution (0.25°×0.25°) ocean subsurface salinity dataset (1–2000 m) for the period 1993–2018, in which the spatial and temporal information (time, longitude, latitude, depth), **previously available the 1°×1° resolution salinity dataset,** and satellite remote sensing data (ADT, SST, SSW) were given as inputs to the FFNN algorithm for reconstruction. By training the functional relationship between input variables and truth values (observed gridded averaged salinity), the reconstruction model was established". With one more sentence (in bold) added.

Second, we also include an analysis to discuss the relative importance of different inputs using Shap-method in the revised manuscript. We find that IAP1° is the most important input, which is physically meaningful because it sets the large-scale pattern of salinity, and the other inputs are indirect information and should play a complementary role (i.e. mainly to add small scale signals).

Finally, we decided to keep "reconstruction" instead of "upscaling", because "reconstruction" is a standard word for derivation of gridded field from observations. In previous temperature/salinity reconstruction efforts, *in situ* observations are firstly gridded into grid averages and then do the mapping (spatial interpolation), for example (Cheng et al. 2017; Cheng et al. 2020; Gaillard et al. 2016; Good et al. 2013; Ishii et al. 2003; Ishii et al. 2017; Levitus et al. 2000; Levitus et al. 2012; Lyman and Johnson 2014; Lyman et al. 2010; Roemmich and Gilson 2009), and also in a recent machine-learning effort (Bagnell and DeVries 2021). You can find more references in our recent review paper in Nature Reviews Earth & Environment (Cheng et al. 2022). Therefore, this is process is a common standard. Doing gridded averages aims to partly homogenize the data and reduce the sub-grid variability.

2 - The "truth" data is made of in situ profiles gridded by simple arithmetic averaging (L143). It is referred to as the "in situ observations" several time in the paper, which I find confusing. Here are three examples :

**Re:** Thanks, we have clarified this issue at several locations.

—> The statement L86 to L89 leads to believe that the present study does use in situ profiles directly as an input as opposed to previous studies : "First, some studies (Lu et al. 2019; Su et al. 2020; Wang et al. 2021) used Argo gridded data rather than in situ salinity observations data as the "truth" to train the machine learning model, and thus the reconstruction error in the Argo gridded data is embedded in the final reconstruction."

**Re:** This is a state of fact because it does not refer to our study, thus we prefer no change made. Because we have clarified this in the following texts, we believe it is sufficient.

—> L94 : "in situ observations" refer to the gridded 0.25x0.25 field here.

**Re:** Revised to "in situ profile observations (processed to a gridded 0.25° × 0.25° arithmetic mean field)"

—> L503 : "truth values (observed salinity)"

**Re:** "truth values (observed salinity)" revised to "truth values (observed gridded averaged salinity)"

Could you please differentiate clearly between the in situ profiles and the "truth" gridded fields of salinity? A gridded field is not "observation".

**Re:** We disagree that "gridded average" is not "observation", it is "observation". Actually, many of the gridded datasets after reconstruction have been commonly regarded as "observation", for example IPCC reports (please check chapter-2 of IPCC-AR6, time series for climate monitoring in WMO State of Climate Report, and BAMS State of Climate report) (Gulev et al. 2021; Johnson et al. 2021).

In addition, we want to clarify an important concept: the term "observation" is not limited to "*in situ* observation", the former has broader context: gridded averages and data products after interpolation are all regarded as "observations" although they are not "in situ observation".

3 - Figure 9 shows the RMSE between IAP0.25, IAP1, Armor3D and EN4 against "observation data" (L355). But is this observation data the "0.25°×0.25°, 1-month, and 41-level grid averages" or the in situ profiles? if it is the grid average, this presentation is biased towards IAP0.25 as it is the only product that was trained with the grid average, so the other products are expected to have higher RMSE.

**Re:** We agree that the comparison is not perfect, because the *in situ* data and gridded averaged fields are all processed by the authors' group, the data used for this validation is not independent. But we don't think the presentation is "biased" because all these products are using essentially the same *in situ* profile database (for example, >99% data in EN4 archive is from WOD, Good et al. 2013). We did use more evaluation approaches to compensate this shortcoming.

And we oppose to compare the final gridded field with *in situ* profiles, because the results will be biased to the regions with more data, especially where there are high-frequency observations (sometimes, there are even >1000 of profiles in one grid box by several moorings). Besides, as the RMSE (between gridded field versus in-situ-profiles) = Reconstruction error + Observational error + Sub-grid variability (<0.25°× 0.25° grid), the results will have loose physical meaning.

If it is the in situ profiles compared to colocated profiles from EN4, Armor3D, IAP0.25 and IAP1, then it would be interesting to see the RMSE between the in situ profiles and the grid average used a "truth" data. Could you add this RMSE profile to the figure?

**Re:** The RMSE between *in situ* profiles and the gridded averages represents the the sub-grid (<0.25°× 0.25° grid) variability, reconstruction error and the observational error. If the purpose is to reconstruct a complete 0.25°× 0.25° field, the sub-grid variability is also "noise" or "error" in the reconstruction (we don't want these variability), thus this RMSE represents the level of noise in different grids. In our reconstruction, we did properly account for these errors (sub-grid variability and observational errors) in reconstruction (see Section 2, uncertainty estimate).

One important concept in data reconstruction field is: "truth" depends on purpose of the reconstruction: it the purpose is to reconstruct a 0.25°× 0.25° resolution dataset, sub-grid (<0.25°) variability is error/noise, thus in situ profiles can't be regarded as "truth" as they have strong sub-grid variability.

Nevertheless, although our gridded average is not perfect because it can't completely remove sub-grid variability, but we do find our gridded averages works better than using *in situ* observations in our global reconstruction.

The IAP1 data is also "interpolated into unified monthly and 0.25°× 0.25°spatial resolution fields" (L131-L132). Could you plot the RMSE between this regridding of IAP1 and in situ profiles on figure 9 too? That would be a good justification for using the neural network for this upscaling.

**Re:** This is a good point and we already had the paper (Fig.2a, c, but for standard deviation). This plot shows the presence of the sub-grid (<1°× 1° grid) variability plus the observational errors.

Best regards,

Etienne Pauthenet

Reference :

Pauthenet, E., Bachelot, L., Balem, K., Maze, G., Tréguier, A. M., Roquet, F., ... & Tandeo, P. (2022). Four-dimensional temperature, salinity and mixed-layer depth in the Gulf Stream,

reconstructed from remote-sensing and in situ observations with neural networks. Ocean Science, 18(4), 1221-1244. **Citation**: https://doi.org/10.5194/essd-2022-236-CC1

**Re:** Thanks for introducing this very interesting paper. We have added this reference in the revised manuscript.

References:

Bagnell, A., and T. DeVries, 2021: 20(th) century cooling of the deep ocean contributed to delayed acceleration of Earth's energy imbalance. *Nat. Comm.*, **12,** 4604.

Cheng, L., K. E. Trenberth, J. Fasullo, T. Boyer, J. Abraham, and J. Zhu, 2017: Improved estimates of ocean heat content from 1960 to 2015. *Sci. Adv.*, **3,** e1601545.

Cheng, L., and Coauthors, 2020: Improved Estimates of Changes in Upper Ocean Salinity and the Hydrological Cycle. *J. Clim.*, **33,** 10357-10381.

Cheng, L., and Coauthors, 2022: Past and future ocean warming. *Nature Reviews Earth & Environment*.

Gaillard, F., T. Reynaud, V. Thierry, N. Kolodziejczyk, and K. von Schuckmann, 2016: In Situ–Based Reanalysis of the Global Ocean Temperature and Salinity with ISAS: Variability of the Heat Content and Steric Height. *J. Clim.*, **29,** 1305-1323.

Good, S. A., M. J. Martin, and N. A. Rayner, 2013: EN4: Quality controlled ocean temperature and salinity profiles and monthly objective analyses with uncertainty estimates. *J. Geophys. Res. Oceans*, **118,** 6704-6716.

Gulev, S., and Coauthors, 2021: Changing state of the climate system. In *climate change 2021: The physical science basis. Contribution of working group I to the sixth assessment report of the intergovernmental panel on climate change* (eds Masson-Delmotte, V. et al.).

Ishii, M., M. Kimoto, and M. Kachi, 2003: Historical Ocean Subsurface Temperature Analysis with Error Estimates. *Mon. Weather Rev.*, **131,** 51-73.

Ishii, M., Y. Fukuda, S. Hirahara, S. Yasui, T. Suzuki, and K. Sato, 2017: Accuracy of Global Upper Ocean Heat Content Estimation Expected from Present Observational Data Sets. *Sola*, **13,** 163-167.

Johnson, G. C., and Coauthors, 2021: Ocean heat content [in State of the Climate in 2020]. *Bull. Am. Meteorol. Soc.*, **102,** S156-S159.

Levitus, S., J. I. Antonov, T. P. Boyer, and C. Stephens, 2000: Warming of the World Ocean. *Science*, **287,** 2225-2229.

Levitus, S., and Coauthors, 2012: World ocean heat content and thermosteric sea level change (0-2000 m), 1955-2010. *Geophys. Res. Lett.*, **39,** L10603.

Lyman, J. M., and G. C. Johnson, 2014: Estimating Global Ocean Heat Content Changes in the Upper 1800 m since 1950 and the Influence of Climatology Choice. *J. Clim.*, **27,** 1945-1957.

Lyman, J. M., and Coauthors, 2010: Robust warming of the global upper ocean. *Nature*, **465,** 334-337.

Roemmich, D., and J. Gilson, 2009: The 2004–2008 mean and annual cycle of temperature, salinity, and steric height in the global ocean from the Argo Program. *Prog. Oceanogr.*, **82,** 81-100.

---

## Author Response (AR1)

**We'd like to thank reviewer#1 for the constructive comments and suggestions. Based on these remarks we have improved the manuscript. Please find below a point-by-point response to each referee's comments. The referee comments are in black, our response is in blue, text changes are in orange.**

Manuscript number: essd-2022-236
Title: Reconstructing ocean subsurface salinity at high resolution using a machine learning approach
MS type: Data description paper
Author(s): Tian Tian, Lijing Cheng, Gongjie Wang, John Abraham, Wangxu Wei, Shihe Ren, Jiang Zhu, Junqiang Song, and Hongze Leng

Submission date for revisions: 4th Oct 2022

Peer-Reviewer #1:
It is a well-written paper on reconstructing subsurface salinity profiles using satellite data and some reanalysis data with the machine learning approach. A very important new contribution to the problems is the development of a better product to examine the vertical structure of the salinity field than ARM and EN4. I have the SST background, so I only have a few minor comments (which I leave to the authors to decide on how to address).

- Define the abbreviation the first time it is used in the text; also, there is no need to define it twice. The author should check it thoroughly. For example, The SST was defined in line 108 but still used in lines 66,84.
Re: Thanks, we have checked all the items and modified them according to this suggestion.

-The authors put the DOI link in the abstract. Whether the link can point to another English version which is http://english.casodc.com/data/metadata-special-detail?id=1546377368443076609 Then more people can use this data
Re: Thanks, we applied for a new DOI link (http://doi.org/10.57760/sciencedb.o00122.00001) which is an English website and more easily accessible to the data. We have updated this link in the revised manuscript.

- This study used the FFNN approach to reconstruct the salinity dataset. Why the authors use this specific method? Have they done the comparison between FFNN-VAR with other models, such as XGBoost, GANs, random fores, for reconstructing the ocean subsurface salinity dataset?
Re: Thanks for pointing this out. The methods have been assessed based on published literatures and our own experiments.
   ● **Published researches.** Stamell et al compared the advantages and disadvantages of three different machine learning methods in reconstructing the global ocean surface $pCO_2$ from sparse observation data, and found that that XGBoost produces the best pCO2 reconstruction overall, but the NN method can be best generalized in poorly sampled regions and time periods. Wang et al. (2021) compared the performance of four machine learning algorithms XGBoost, MLR, RF and NN in estimating the subsurface temperature in the western Pacific, and proved that the neural network model outperformed the other three machine learning models. Lu et al. (2019) estimated subsurface temperature using cluster-neural network method, and showed that this method was superior to clustering linear

regression and random forest method. The above mentioned research shows that the NN method has superior generalization ability and is more robust for ocean data reconstruction.

● **Our experiments.** We have used "synthetic data" approach to test the performance of different ML. CNRM-CM6-1 high-resolution model simulation is used (historical simulation that includes all climate forcings and is part of CMIP6). Because model results are with global coverage, dynamically consistent, thus can be used as **"true"** of salinity. We resample the model data according to the location of *in situ* observation to construct the **"synthetic observations"**. The synthetic observations are prepared in two counterparts. The first is after further perturbation by observational errors/noises (denoted as **"perturbed"** data to account for the impact of observational errors). The observational errors are specified in section 2.3.3. and Fig.2. The second is no perturbation (so the only error source of reconstruction is data sampling, this data is denoted as "**non-perturbed**"). Based on these synthetic data, we test different reconstruction schemes and compare the applicability of FFNN and LightGBM (an improved method of XGBoost) to reconstruct globally ocean salinity data from sparse data. Fig. X1 and Fig.X2 show that the FFNN exhibits the lower RMSE (~0.035 psu) and the higher correlation coefficient (CC) (~0.866) compared with LightGBM for non-perturbed synthetic data. The perturbed data shows consistent results, although the addition of noises led to slightly higher RMSE and lower CC for both methods. In particular, perturbation of data induced a CC degradation of 3.5% and 6.4% for FFNN and LightGBM, respectively (Fig.X2d). Thus, the addition of observational noises lead to larger performance degradation for LightGBM than FFNN, suggesting that FFNN is more robust.

Finally, we also compare the performance of FFNN and LightGBM in reconstructing salinity using FFNN. The results show that the salinity field reconstructed by LightGBM had many non-continuous stripe-like structures (Fig.X3), which is apparently non-physical. This is associated with the intrinsic property of the LightGBM approach: 1) LightGBM discretize continuous variable into small bins by splitting the tree nodes; 2) Tree based models give stripe-like predictions as the model was trained using spatially sparse data.

With those tests and considerations, we found FFNN be an optimal choice to reconstruct subsurface salinity data in this study.

We have summarized the above analyses into the supplementary material and add several sentences in the main manuscript to avoid the overlength of the main manuscript. "We chose FFNN because it has been shown to be superior to the other three widely used machine learning approaches in reconstructing ocean parameters. Our own evaluation based on synthetic data and salinity observations (see supplementary material) also reveals that FFNN is a robust approach and leads to the smallest error compared with other approaches [e.g., light gradient boosting machine (LightGBM)]."

[Figure]

Figure X1: Distribution of 1–2000 m averaged RMSE and correlation coefficient for different ML approaches. Blue markers represent the perturbed data; Red markers represent the non-perturbed data.

[Figure]

Figure X2: Statistical metrics for FFNN and LightGBM methods. (a) 1–2000 m averaged RMSE. (b) Increase of RMSE from the non-perturbed data to the perturbed data; (c) 1–2000 m averaged correlation coefficient (CC). (d) Degradation of CC from the non-perturbed data to the perturbed data.

[Figure]

Figure X3: The geographical distribution of salinity anomalies in the Kuroshio and Gulf Stream region from data reconstructed by FFNN (left) and LightGBM (right) method in January 2016: (a–b) Northwest Pacific region; (e–f) Northwest Atlantic region.

- Figure 11 shows the seasonal fluctuation of RMSE of IAP 0.25 AND IAP 1. It's better to add the seasonal fluctuations of other in situ and reanalysis products to compare their performance.

Re: Thanks. According to your suggestion, we made some additional analysis using one representative reanalysis product: ORAS4 and one representative objective analysis product: EN4 data. We found that RMSE has strong seasonal fluctuations consistent with IAP1 and IAP0.25° (Fig.X4). We have tried to avoid including more datasets because a thorough inter-comparison of available products should be done in a separate study with carefully designed metrics. Therefore, our analyses only serve to indicate the robustness of our results.

[Figure]

Figure X4: The 1–2000 m average RMSE time series for the IAP0.25°, IAP1°, ORAS4 and EN4 datasets for the globe and three ocean basins from 1993 to 2018.

- The authors should analysis the errors with respect to the input (SLA, SSTA, SSSA, UWSA, VWSA): Which input contributes the most errors? Which input could be the primary surface independent parameter for estimating salinity?

Re: Thanks, this is a great point and definitely help to better understand the approach. In the revised manuscript, we have used an approach named SHAP (SHAPley Additive exPlanations) to evaluate the relative contribution of different input parameters. SHAP is useful in explaining the various supervised learning models and assigns an importance value for each input variable for a specific prediction.

The analysis is supplemented in the section 2.3.4 and section 6.

**"2.3.4 Evaluating the relative importance of different inputs**

[revised manuscript text omitted]

**We'd like to thank reviewer#2 for the constructive comments and suggestions. Based on these remarks we have improved the manuscript. Please find below a point-by-point response to each referee's comments. The referee comments are in black, our response is in blue, text changes are in orange.**

Manuscript number: essd-2022-236
MS type: Data description paper
Title: Reconstructing ocean subsurface salinity at high resolution using a machine learning approach
Author(s): Tian Tian, Lijing Cheng, Gongjie Wang, John Abraham, Wangxu Wei, Shihe Ren, Jiang Zhu, Junqiang Song, and Hongze Leng
Submission date for revisions: 4th Oct 2022

Peer-Reviewer #2:

In this work the authors describe a new product of global subsurface salinity at a high resolution (0.25°x0.25°) covering 41 vertical levels (in the range 1-2000m), named IAP0.25°.
This product is obtained using a Feed Forward Neural Network model designed by the authors, which is trained from several input features taken from satellite sets and reanalysis to reconstruct the subsurface salinity product. Interpolated in situ observations of salinity profiles has been considered as ground truth to compare reconstruction findings. Finally, IAP0.25° is evaluated by comparison with three independent ocean products of salinity.

The overall presentation of the new product, results and evaluation metrics is of high quality, hence in my opinion the manuscript should be considered positively for publication in ESSD. I would suggest a minor revision, mainly because I find that improvements in presentation and some additional comments have to be considered. I give also line by line suggestions to ease revision.

- The paper is quite long: I find it is possible to shorten some parts by avoiding repetitive sentences or being more concise and direct in some subsections. This is particularly true, for example, for the Introduction. I believe that shortening the manuscript would aid the overall readability, hence facilitating the choice of using the authors' dataset.
Re: Thank you for your guidance and suggestions. We have revised the manuscript based on your constructive and helpful suggestions. The abstract has been cut and many repeated discussions in the manuscript have been removed.

- The link of the IAP0.25° DOI should be given for the English version of the website page. Indeed you end up in the Chinese one and when you switch to English you are brought back to the home page, which is confusing. In the website the dataset is told to cover 0-2000m instead of 1-2000m (which is correctly reported in the manuscript and seen from the netcdf downloaded)
Re: This is a good point. According to your suggestion, we applied for a new DOI link (http://doi.org/10.57760/sciencedb.o00122.00001) which is an English version and more easily accessible to the data. We have updated this link in the revised manuscript. We have also replaced "0-2000m" with "1-2000m" in the manuscript.

- The FFNN model description needs some additional details. In general the authors have fully described data and reconstruction and evaluation, but less attention has been paid in motivating the NN choice and structure. I believe that adding this kind of comments would aid in understanding the background ratio.

Re: Thanks for pointing this out. The methods have been assessed based on published literatures and our own experiments.

- **Published researches.** Pauthenet et al compared the advantages and disadvantages of three different machine learning methods in reconstructing the global ocean surface $pCO_2$ from sparse observation data, and found that that XGBoost produces the best pCO2 reconstruction overall, but the NN method can be best generalized in poorly sampled regions and time periods. Wang et al compared the performance of four machine learning algorithms XGBoost, MLR, RF and NN in estimating the subsurface temperature in the western Pacific, and proved that the neural network model outperformed the other three machine learning models. Lu et al estimated subsurface temperature using cluster-neural network method, and showed that this method was superior to clustering linear regression and random forest method. The above mentioned research shows that the NN method has superior generalization ability and is more robust for ocean data reconstruction.

- **Our experiments.** We have used "synthetic data" approach to test the performance of different ML. CNRM-CM6-1 high-resolution model simulation is used (historical simulation that includes all climate forcings and is part of CMIP6). Because model results are with global coverage, dynamically consistent, thus can be used as **"true"** of salinity. We resample the model data according to the location of *in situ* observation to construct the **"synthetic observations"**. The synthetic observations are prepared in two counterparts. The first is after further perturbation by observational errors/noises (denoted as **"perturbed"** data to account for the impact of observational errors). The observational errors are specified in section 2.3.3. and Fig.2. The second is no perturbation (so the only error source of reconstruction is data sampling, this data is denoted as "**non-perturbed**"). Based on these synthetic data, we test different reconstruction schemes and compare the applicability of FFNN and LightGBM (an improved method of XGBoost) to reconstruct globally ocean salinity data from sparse data. Fig. X1 and Fig.X2 show that the FFNN exhibits the lower RMSE (~0.035 psu) and the higher correlation coefficient (CC) (~0.866) compared with LightGBM for non-perturbed synthetic data. The perturbed data shows consistent results, although the addition of noises led to slightly higher RMSE and lower CC for both methods. In particular, perturbation of data induced a CC degradation of 3.5% and 6.4% for FFNN and LightGBM, respectively (Fig.X2d). Thus, the addition of observational noises lead to larger performance degradation for LightGBM than FFNN, suggesting that FFNN is more robust.

Finally, we also compare the performance of FFNN and LightGBM in reconstructing salinity using FFNN. The results show that the salinity field reconstructed by LightGBM had many non-continuous stripe-like structures (Fig.X3), which is apparently non-physical. This is associated with the intrinsic property of the LightGBM approach: 1) LightGBM discretize continuous variable into small bins by splitting the tree nodes; 2) Tree based models give stripe-like predictions as the model was trained using spatially sparse data.

With those tests and considerations, we found FFNN be an optimal choice to reconstruct subsurface salinity data in this study.

We have summarized the above analyses into the supplementary material and add several sentences in the main manuscript to avoid the overlength of the main manuscript. "We choose FFNN because it has been shown to be superior to the other three widely used machine learning approaches in reconstructing

ocean parameters. Our own evaluation based on synthetic data and salinity observations (see supplementary material) also reveals that FFNN is a robust approach and leads to the smallest error compared with other approaches [e.g., light gradient boosting machine (LightGBM)]."

[Figure]

Figure X1: Distribution of 1–2000 m averaged RMSE and correlation coefficient for different ML approaches. Blue markers represent the perturbed data; Red markers represent the non-perturbed data.

[Figure]

Figure X2: Statistical metrics for FFNN and LightGBM methods. (a) 1–2000 m averaged RMSE. (b) Increase of RMSE from the non-perturbed data to the perturbed data; (c) 1–2000 m averaged correlation coefficient (CC). (d) Degradation of CC from the non-perturbed data to the perturbed data.

[Figure]

Figure X3: The geographical distribution of salinity anomalies in the Kuroshio and Gulf Stream region from data reconstructed by FFNN (left) and LightGBM (right) method in January 2016: (a–b) Northwest Pacific region; (e–f) Northwest Atlantic region.

- Understanding which input mostly influences the salinity reconstruction and causes greater propagation error would be very interesting in this work, and in my opinion would enhance completeness of the presentation (this is stated as a future step in the Summary section, hence to be considered only as a suggestion).

Re: Thanks, this is a great point and definitely help to better understand the approach. In the revised manuscript, we have used an approach named SHAP (SHAPley Additive exPlanations) to evaluate the relative contribution of different input parameters. SHAP is useful in explaining the various supervised learning models and assigns an importance value for each input variable for a specific prediction.

The analysis is supplemented in the section 2.3.4 and section 6.

**"2.3.4 Evaluating the relative importance of different inputs**

[revised manuscript text omitted]

- In several points some statements sound very vague or too general (see line by line comments hereafter). I suggest to be more precise. For example, the authors could revise how they refer to machine learning in a vague way: it might be more interesting to focus on neural networks only, since this is the model choice for this study.
Re: We appreciate for your detailed suggestions, we have revised the manuscript based on your suggestions, as introduced below.

1. Introduction
- (54) typo *below the ocean surface
Re: Corrected.

- (62) what to you mean with sufficient data quality? This is vague

Re: Great suggestion, sentence revised to "high-quality observational datasets with resolutions higher than 1° × 1°  could be useful for ocean and climate research"

- (75-76) state better the limitations

Re: Great suggestion, Sentence revised to "However,  both dynamic and statistical approaches have obvious limitations. The dynamic approach is overly dependent on physical assumptions, which are always simplified such as relying on Surface Quasi-Geostrophic dynamics to derive subsurface signals from surface changes. Caveats of the statistical approach is the lack of physical constraints and the simplified assumptions. For example, some approaches have simplified the nonlinear relationship between surface dynamic height and subsurface temperature/salinity to a linear relationship."

- (77-78) in this study you inspect NN for reconstruction of salinity field only, not for other ocean subsurface fields. state better

Re: Thanks a lot. Following your next suggestion, we have removed this sentence. ""

- (94-96) a bit repetitive with (77-79). revise by shortening

Re: Thanks, we removed the sentence ""

- (86) I would not talk about "major deficiencies", which sounds too strong for the subsequent comments. Maybe *some differences, *some limitations or similar

Re: Great suggestion, Sentence revised to "Although these studies provide some hints that machine learning approaches can be useful in data reconstruction applications,  there are still some limitations."

- (97) the second objective introduced does not sound as an objective

Re: This sentence revised to "Second, the new machine-learning-based high-resolution (0.25° × 0.25°) salinity dataset will be comprehensively evaluated in this study, which facilitate its further applications"

- (103) currently your section 6 is of Data availability (this could become your last section 7, probably more appropriate)

Re: Sentence modified to "Importance of each feature for the reconstruction is described in section 6. Data availability is described in Section 7. The results of the study are summarized and discussed in section 8."

2. Data & Methods

- (107-109) this should be said in the introduction, not here. I would not say "a machine learning model", which sounds very general; instead point out it is your model- this happens also in other points of the paper.

Re: Thanks. We put this part of the statement in the penultimate paragraph of the introduction. "This paper explores the feed-forward neural network (FFNN) approach to reconstruct a high-resolution (processed to a gridded 0.25° × 0.25° arithmetic mean field in this study, detailed in the following text) ocean subsurface (1–2000 m) salinity dataset for the period 1993–2018 by merging *in situ* profile observations with high-resolution (0.25° × 0.25°) satellite remote sensing altimetry absolute dynamic topography (ADT), SST, sea surface wind (SSW) data, which included zonal (USSW) and meridional (VSSW) components, and a coarse resolution IAP1° gridded salinity product."

"a machine learning model" is changed to "FFNN".

- (110, 115, 121) sentences "data were from.." could be improved with expressions as: we use, we downloaded, data was extracted from...

Re: Great suggestion. We change "data were from" to "data was extracted from".

- (112) sounds like you have done the optimal interpolation

Re: We have removed this sentence "".

- (116) take off "and extrapolating"

Re: Done

- (119-120) take off last sentence, you say this at the end of the 2.1.1 ssec, repetitive

Re: Done

- Table 1. Should be better organised, I suggest you should have: Data type, Variable, Dataset, Data Source, Horizontal resolution, Vertical coverage and resolution, Time period, Reference, DOI; hence correct information given (e.g., for SST you would have Variable: SST, Dataset: OISST, Data Source: NOAA, etc.). Pay attention to classifying Salinity observations as Input, since indeed you use it as ground truth to which you compare the model output.

Re: Great suggestion, we have revised the Table 1 following your suggestion. Note that for some data (such as IAP1°), there is no DOI, just a website and a paper.

| Data type | Variable | Dataset | Data source | Horizontal resolution | Vertical coverage and resolution | Time period | Reference | DOI/URL |
|---|---|---|---|---|---|---|---|---|
| Input | ADT | CMEMS | CMEMS | 0.25° × 0.25° | Sea surface | 1993–2020 | (Mertz et al., 2016) | https://doi.org/10.48670/moi-00148 |
| Input | SST | OISST | NOAA | 0.25° × 0.25° | Sea surface | 1981–2022 | (Huang et al., 2021) | https://www.ncei.noaa.gov/products/optimum-interpolation-sst |
| Input | SSW | CCMP | NCAR | 0.25° × 0.25° | Sea surface | 1987–2019 | (Wentz et al., 2016) | https://doi.org/10.5065/4TSY-K140 / |
| Input | Salinity | IAP1° | IAP | 1° × 1° | 41 levels (1–2000 m) | 1960–2021 | (Cheng and Zhu, 2016) | http://www.ocean.iap.ac.cn/ |

| | | | | | | | | |
|---|---|---|---|---|---|---|---|---|
| Input | Salinity observations | *In situ* observations | WOD | Averaged into 0.25° × 0.25° | Interpolated to 41 levels (1–2000 m) | 1960–2021 | (Boyer et al., 2018) | https://www.ncei.noaa.gov/products/world-ocean-database |
| Validation | Salinity | ARMOR3D | CMEMS | 0.25° × 0.25° | 50 levels (1–5000 m) | 1993–2020 | (Mertz et al., 2016) | https ://doi.org/10.48670/moi-00052 |
| Validation | SSS | SMAP | NOAA | 0.25° × 0.25° | Sea surface | 2015–2019 | (Vinogradova et al., 2019) | https://data.remss.com/smap/SSS/V04.0/ |
| Validation | Salinity | EN4 | UK Met Office | 1° × 1° | 42 levels (1–5500 m) | 1940–2018 | (Gouretski and Reseghetti, 2010) | https://www.metoffice.gov.uk/hadobs/en4/ |

- Eventually insert DOIs in text for each product for completeness

Re: Thanks a lot. I texted the DOI or url (if DOI is not available) of the data source in table 1 to avoid the overlength of the manuscript.

- (131) say something more of interpolation technique adopted

Re: Great suggestion, Sentence revised to "All of the above-mentioned products were processed into monthly averages, and IAP1° data were linearly interpolated to unified 0.25° × 0.25° resolution fields , which were used as inputs for  the FFNN approach."

- (139-140) state better, e.g., anomaly profiles are derived by subtracting monthly climatologies from salinity profiles.

Re: Great suggestion. Sentence revised to "and anomaly profiles are derived by subtracting monthly climatologies from salinity profiles ."

- (140-141) not clear to me, to state better

Re: The sentence has been changed to "Finally, the salinity anomalies were averaged into 0.25° × 0.25°, 1-month, and 41-level grid boxes via simple arithmetic averaging, without spatial interpolation and smoothing. The reconstructions are applied to the anomaly fields, similar to previous objective analyses, because of the larger spatial decorrelation length scale of the anomaly fields than the absolute fields (i.e., Levitus et al. 2009; Cheng et al. 2017). The gridded averages used in this study are also consistent with most previous studies to reduce the sub-grid variability and observational noises.".

- why do you say that SSA were averaged in 0.25 resolution? Did you mean interpolated?

Re: This is just averaging all anomalies into 0.25° × 0.25°, 1-month, and 41-level grid boxes. No interpolation is applied, so there are many grids without data (gaps).

- (146) "certain tests" is too vague. Say if you tested your method on raw profiles and found no differences, or say something more about the tests (without needing to show figures or results about)

Re: To avoid confusion, we have removed this sentence.

- (150) remove "including", since you are stating all sets

Re: Done

- (156-157) remove "which have spatial resolution of 0.25", repetitive

Re: Done

- (169) You should start this subsec by describing the FFNN, not the data. These first sentences could be moved in data subsec or later on.
Re: We moved the first sentence before "The structure of the FFNN used in this study is illustrated in Fig. 1" because the derivation of anomaly fields have already introduced in the Method section.

- (186-187) complexity of a NN depends not only on how many neurons or layers are chosen, but also on type of layers and activation functions.
Re: Great suggestion, Sentence revised to "The complexity of the FFNN depends not only on how many neurons or layers are chosen, but also on type of layers and activation functions. "

- (190-192) make it shorter and less repetitive
Re: Great suggestion, Sentence revised to "The input x includes longitude, latitude, depth, time, IAP1SA, ADTA, SSTA, USSWA, VSSWA."

- (195) you should not refer to your model generically as "a FFNN" but "the FFNN"
Re: Done

- (196) take off "based on training set" uninformative
Re: Done

-- (199) I would not talk about "reconstruction effect"
Re: Sentence revised to "A grid search strategy (Liashchynskyi et al. 2019) was used to optimize the structure of the neural network. The optimized neural network we used consists  of one input layer, one output layer, and four hidden layers; the number of neurons in each hidden layer was set to 256, 128, 64, and 32; the activation function was the Rectified Linear Unit; the optimizer was the root mean square propagation (RMSProp); the learning rate was 0.001;"

-- Figure 1. when showing the output information you give several maps of subsurface salinity anomalies, since your method is reconstructing each time step separately from the others I suggest you leave only one map also in the figure. Caption: I suggest you change in "Schematics of the FFNN architecture for subsurface salinity anomalies reconstruction (IAP0.25)"
Re: Great suggestion. We have revised the figure as suggested.

- which is your cost function minimized for training? did you use a GD algorithm? say more
Re: The cost function minimized for training as follows:

$$J(\theta_0, \ \theta_1, \ \ldots \theta_n) = \frac{1}{2m} \sum_{i=1}^{m} (h_\theta(\mathbf{X}^{(i)}) - y^{(i)})^2$$

Where **X** = (longitude, latitude, depth, time, IAP1SA, ADTA, SSTA, USSWA, VSSWA), y is the "truth values", $m$ is the number of samples, c is the FFNN model for training, $\theta$ is the parameter of the model, and the training objective is the minimum J.

We used the Root mean square propagation (RMSProp) optimizer, which was widely used for the stochastic problem. RMSProp is the optimization machine learning algorithm to train the Neural Network by different adaptive learning rate and derived from the concepts of gradients descent and RProp. Combining averaging over mini-batches, efficiency, and the gradients over successive mini-batches, RMSProp can reach the faster convergence rate than SGD, Momentum, and NAG.

Sentence revised to "the activation function was the Rectified Linear Unit; the optimizer was the root mean square propagation (RMSProp); the learning rate was 0.001; the cost function minimized for training as follows:

$$J(\theta_0, \ \theta_1, \ ...\theta_n) = \frac{1}{2m}\sum_{i=1}^{m}(h_\theta(\mathbf{X}^{(i)}) - y^{(i)})^2$$

where X = (longitude, latitude, depth, time, IAP1SA, ADTA, SSTA, USSWA, VSSWA), y is the "truth values", $m$ is the number of samples, $h_\theta$ is the FFNN model for training, $\theta$ is the parameter of the model, and the training objective is the minimum J."

- (209) I would mention that the independent dataset are those introduced in ssec 2.2
Re: Thanks, but this dataset is different from section 2.2. In section 2.2, we introduced some gridded datasets constructed by independent international groups. Here the "independent data" means we have withheld 20% of data that is not used in training, so these data can be used as independent data to test the approach.
Sentence revised to "To evaluate the reconstruction using independent test data, a 5-fold cross validation approach was used."

- (243-244) is variance the standard deviation? usually variance=std**2
Re: Thanks for spotting this, we agree that using "Variance" might be misleading. We have changed "Variance" to "Var" to represent the "variability", because it is a measure of subgrid variability (<0.25° and <1-month).

- (256) I would add "estimate machine learning methods uncertainty"
Re: Great suggestion. Sentence revised to "This is one of the most popular ways to estimate machine learning **method** uncertainty".

-- (263) missing ending dot
Re: Modified

--Figure 2. (b&d) it seems to me that there might have been higher values than 0.2, flattened to be exactly 0.2. if this is the case please adapt colorbar with pointed end for out-of-range values. Eventually it would be interesting to have the four panels shown with same colorbar limits to be able to compare effectively two different depths' results (if graphically informative)
Re: We have modified the colorbar to the ones with pointed end, to better illustrate the higher values. And also, we used the same colorbar limits for all panels for a better intercomparison.

3. reconstruction Results
-- (275) typo: take of "as"
Re: Corrected.

- (276) you should quantify the consistency between your reconstruction and IAP1 & ARMOR3D with some metric
Re: Thanks. The quantification of the consistency between IAP0.25, IAP1 and ARMOR3D is mainly done in Section 3.3. Sections 3.1 and 3.2 are only intended to evaluate the overall performance of IAP0.25 on spatial and vertical structures with a few illustrative examples.

- (280) take off "indicating greater resolution", uninformative. Next sentence take off "of change", the patterns you are showing are of one particular month.
Re: Done

- (282) How did you perform subtraction of IAP0.25 and IAP1? Did you first have IAP1 interpolated?
Re: Yes, IAP1° data were linearly interpolated to 0.25° × 0.25° resolution fields, and the impact of interpolation method is negligible (it makes sense because there is no change within 1° × 1° grid for IAP1°).

- I find it curious that you first show results of 5m and 100m depth globally, but then you turn to 100m & 300m for specific regions. Why is this? if there is no reason I would try to be consistent between global and detail analysis
Re: Great suggestion. We did show this just arbitrarily, it does not change the key message for different choices. But for consistency, we have replaced the global results with 100 and 300 m in the revised manuscript.

- Figure 3 & 4. Refer in caption to each product by giving subfigures letters. You don't mention comparison between IAP025 and IAP1 in the caption (panel e that could be given close to the two products images)
Re: Thanks. We have revised the figure accordingly. Caption revised to "Spatial distribution of salinity anomalies  of (a) IAP0.25°, (b) IAP1°, (c) ARMOR3D,  (d) in situ observations, and (e) IAP1° minus IAP0.25° at 100 m, as well as the spatial distribution of (f) ADTA and (g) SSTA in January 2016"

- (298) You talk about reliability of the reconstruction by giving very qualitative comments, this should be quantified for robustness via some metric
Re: Thanks. The quantification of robustness is mainly done in Section 4. Here we only discuss the robustness of the reconstruction method qualitatively for some examples, we believe it helps the audience to gain a better illustrative impression before any statistical results are shown.

- (327) typo: Figure 2b and *d
Re: Thanks for spotting this. After checking, it should be Figure 7b and d, which have been corrected.

- (334) It is not clear to me when you state that ADT change should correspond better with subsurface salinity. I would suggest to state the whole ADT paragraph better
Re: Thanks, we should be clarified that ADT change should correspond better with thermocline change (first baroclinic mode), thus the salinity change near the thermocline should resemble the ADT in many

places. These sentences have been modified to "To the first order, in the thermocline regions, the ADT change should correspond better with thermocline change (first baroclinic mode), thus the salinity change near the thermocline should resemble the ADT in many places. This is supported by Fig. 7 (red line in Fig. 7a vs. 7d); for example, the large positive anomaly near 48°W, 60°W, and 68°W revealed by both the *in situ* salinity profile and ADT data. This indicates a positive contribution of ADT to the reconstruction."

- (345) take off "certain", it sounds vague
Re: Done

- Figure 7a. Legend should be put all together possibly internal to the image. I suggest to put longitude coordinates on the top of the figure to aid readability of comments.
Re: Great suggestion, we have revised the figure as suggested.

- (359) "seems that" does not sound as a statement. Substitute "in the global area" with "At a global scale"
Re: Great suggestion, Sentence revised to "First,  IAP0.25_RMSE is smaller than all other data products for both a global- and basinal averages. At a global scale , the maximum values are 0.37 psu for IAP0.25_RMSE, 0.50 psu for IAP1_RMSE, 0.55 psu for ARM_RMSE, and 0.56 psu for EN4_RMSE near the sea surface."

- (363-365) very heavy to read, try to improve
Re: Great suggestion, Sentence revised to " Globally, the 1–2000 m mean IAP0.25_RMSE is 0.016 psu, 0.019 psu and 0.021 psu lower than IAP1_RMSE, ARM_RMSE and EN4_RMSE, respectively (Table 2)."

- Figure 9. The best way for comparing graphically rmse over these regions is to have same figure limits for the upper panels (same for the lower panels).
Re: Thanks a lot. I tried to set the x-axis of the subgraph to the same limit, which might cause the curves in the graph clustering together and less visible (Fig.X5 b.d.f.h). So, in order to better illustrate the difference between the curves, we decided to keep the original figure.

[Figure]

**Figure X5: (a–d) Vertical distribution of RMSE for IAP1°, ARMOR3D, EN4, and IAP0.25° for the globe and three major basin regions during 1993–2018. (e–h) Vertical distribution of RMSE for IAP1°, ARMOR3D, and EN4 minus IAP0.25°, respectively.**

- (395) take off parenthesis. why did you include lower quality data?

Re: We decided to remove "".

- (397) I would say even something about lowest reduction

Re: One sentence added "The lowest RMSE reduction is seen in the Indian Ocean, likely associated with smaller area of the western boundary current systems and relatively less meso-scale activities compared with the other two basins."

- Figure 11. The y-axis limits should be the same for all subfigures for a more informative graphical comparison of results. In the caption I would mention that time series are averaged over the different areas

Re: Great suggestion, we have revised the figure as suggested.

- (404) *reveals, instead of suggests, more precise

Re: Done

- (406) *with the other products considered

Re: Done

4. Five-fold cross validation and uncertainty estimate

- (412) I would recall these independent observations, referring also to subsec 2.2.

Re: Thanks a lot. Here, independent observations refer to test data, not independent data in subsec 2.2.

- (421) typo: * are obviously not biased
Re: Done

- (424-435) it is confusing how you comment the findings. First you talk about Figure 12e, then 12a-12d and then 12e again. Revise the whole paragraph, possibly shortening
Figure 12. in the caption the sentence on correlation coefficient should be found when talking about panels a-d
Re: This paragraph has been revised to "Besides, we calculated the RMSE and its degradation between the reconstructed salinity fields and in situ observations of the training and testing sets at each depth layer (Fig. 12). The degradation rate is defined as: (RMSE of the testing set - RMSE of the training set) / (RMSE of the training set), to quantify the generalization of the model. Fig. 12(a) shows that RMSE of the testing set is consistent (only marginally higher than) with that of the training set. The degradation rate decreases rapidly with depth, about 5.49% at the surface and 0.10% at 100 m. Specifically, at 10 m, the RMSE is 0.261 psu for the training set and 0.269 psu for the testing set, the degradation rate is 3% (Fig. 12b, c); and at 100 m, the RMSE is decreases from 0.1403 psu (training set) to 0.1401 psu (testing set) (Fig. 12d, e). Besides, the correlation coefficient is slightly lower for the testing set: for example, 0.686 at 10 m but 0.707 for training set at the same depth; at 100 m, it decreases from 0.625 (training) to 0.623 (testing). As the testing set is independent from training set, this test indicates that the FFNN model does not experience serious overfitting, and the method is valid."

Caption revised to "Figure 12: (a) Vertical distribution of RMSE for the training set and testing set for global region (bottom x-axis) and the RMSE degradation from the training set to the testing set (top x-axis). (b–e) Density distribution diagrams at depths of 10 m and 100 m for the training set and testing set, as well as RMSE and correlation coefficients (r) for the corresponding layers.  The color-coded blocks represent the density of samples."

5. Evaluation of the major climatic patterns
- (454) saying a "very significant linear trend" is too vague. Have you computed it with a p-value? what is the annual rate of change? state better
Re: We have removed "very significant", instead, give the linear trend with 90% CI to indicate the uncertainty (the error bar is calculated taking account of the reduction of degree of freedom). Sentence revised to "The global-scale SC2000 increases significantly from 1993 to 2018, with a  linear trend of $0.045 \pm 0.0058$ psu century$^{-1}$ (at 90% confidence level, the reduction of degree of freedom has been accounted for in this calculation)."

- Figure 13. the title shoud be Salinity Contrast index at Surface/0-2000m. I believe no y label is needed, and anyways should not be salinity since SC is shown
Re: Thanks, we have revised the figure as suggested.

- (461) *for both the salinity at surface and averaged over the 0-2000m volume

Re: Sentence revised to "Besides the global-scale SC metric, we also present the  time series over the globe and in different ocean basins for IAP1° and IAP0.25° from 1993 to 2018 for both the salinity anomalies at the surface (S0)  and averaged over the 1–2000 m volume (S2000) in Figs. 14 and 15.".

- (475) To me the sentence regarding increased SC is not clear

Re: This sentence was not correct. Revised to "The contrast salinity change between Pacific Ocean (decreasing, Fig. 15b) and Atlantic Ocean (increasing, Fig. 15c) is associated with increased inter-basin transport of water vapor from the Atlantic to the Pacific (Reagan et al., 2018; Curry et al., 2003)".

- (479) you are talking about anomalies, why should average values impact the change? state better

Re: Thanks, the sentence has been modified to "S2000 increases in the North Indian Ocean (Fig. 15e) but decreases in the South Indian Ocean (Fig. 15f), showing a "salty gets saltier, fresh gets fresher" change, mainly because of the amplified global hydrological cycle.".

- (486) typo: *systematic twice

Re: Done.

- Figure 14 & 15. if possible all subfigures should have same y axis limits to ease comparison. Are these salinity anomalies?

Re: Accordingly, we have tried to set the Y-axis of the panels to the same limit, but the curves in the graph loos too flat and many variabilities are less visible (Fig.X6), so we still keep the original y limits. Nevertheless, we made some improvements to the graph, such as changing the titles to S0 and S2000.

[Figure]

**Figure X6: The global (a) and basinal (b-f) salinity anomalies (SA) time series at surface from 1993 to 2018 for IAP1° and IAP0.25° respectively. Both monthly and 12-month running smoothed time series are presented, all data are relative to a 1993–2015 baseline.**

6. Data availability
- (494) take off *mainly
Re: Done

- (498) *at a monthly resolution
Re: Done

7. Summary and Discussion
- (502) *were given as inputs to the FFNN algorithm for reconstruction
Re: Done

- (508) it is vague to say that IAP025 performs best compared with many available gridded products. state better
Re: Revised to "(1) IAP0.25° salinity data maintain the large-scale information from IAP1° gridded data. Because previous evaluations suggest that IAP1° provides more physically tenable large-scale patterns and long-term climate change and variabilities compared to many available datasets, thus the reliability of large-scale signals also becomes an advantage of the new IAP0.25° product.".

- (510) you should mention that for point (2) you are referring to subsurface salinity field. Indeed I don't see why separating this point from previous one. Instead, I would say something of your findings of rmse or climatic patterns, which are not cited in this summary

- I suggest that this section should be revised trying to make it more appealing, and pointing more clearly to your findings.

Re: Thanks a lot. In view of the above three suggestions, summary revised to "(1) IAP0.25° salinity data maintain the large-scale information from IAP1° gridded data. Because previous evaluations suggest that IAP1° provides more physically tenable large-scale patterns and long-term climate change and variabilities compared to many available datasets, thus the reliability of large-scale signals also becomes an advantage of the new IAP0.25° product. (2) Compared with IAP1°, the RMSE of IAP0.25° can be reduced by ~11% on a global average. Besides, IAP0.25° shows more realistic spatial signals in the Gulf Stream, Kuroshio, and Antarctic Circumpolar Current regions with strong mesoscale variations than the IAP1°product, indicating that FFNN can effectively transfer small-scale spatial variations in ADT, SST and SSW fields into the 0.25° × 0.25° salinity field. It thus serves as an improvement on the currently available IAP data. (3) We show that the FFNN approach is effective in merging different kinds of Earth observations, and the method is robust and can be reliably used for ocean state reconstruction, thus can complement the existing data assimilation and objective analysis methods.".

---

## Author Response (AR2)

October 27, 2022

**Manuscript number**: essd-2022-236

**MS type**: Data description paper

**Title**: Reconstructing Ocean subsurface salinity at high resolution using a machine learning approach

**Authors**: Tian Tian, Lijing Cheng, Gongjie Wang, John Abraham, Wangxu Wei, Shihe Ren, Jiang Zhu, Junqiang Song, Hongze Leng

Dear Dr. Salvatore Marullo and Dr. Giuseppe M.R. Manzella,

Thanks very much for your hard work toward the publication of our manuscript.

In the last round of review, we have added Mr. Wangxu Wei as a new co-author. During the last review, both reviewers suggest adding a new analysis to quantify the relative importance of different inputs. Therefore, we have brought Wangxu Wei in our group, who worked on the "Shap" method and the related analyses in the revised manuscript. That's why we have added his name in the author list during the last revision (submitted at Oct.4). We also adjusted the "Author Contributions" section of the manuscript accordingly.

We would be grateful if you could give it a consideration and approve this addition. Thanks in advance.

Regards

Lijing Cheng, on behalf of all co-authors
Institute of Atmospheric Physics
Chinese Academy of Sciences
Beijing, 100029, PR, China
chenglij@mail.iap.ac.cn